# Extended DNA threading through a dual-engine motor module of the activating signal co-integrator 1 complex

Junqiao Jia [1,10], Tarek Hilal [1,2], Katherine E. Bohnsack [3], Aleksandar Chernev[4], Ning Tsao[5], Juliane Bethmann[4,6], Aruna Arumugam [1], Lane Parmely[5], Nicole Holton[1], Bernhard Loll [1], Nima Mosammaparast [5], Markus T. Bohnsack[3,7,8], Henning Urlaub [4,6] & Markus C. Wahl [1,9] ✉

Activating signal co-integrator 1 complex (ASCC) subunit 3 (ASCC3) supports diverse genome maintenance and gene expression processes, and contains tandem Ski2-like NTPase/helicase cassettes crucial for these functions. Presently, the molecular mechanisms underlying ASCC3 helicase activity and regulation remain unresolved. We present cryogenic electron microscopy, DNA-protein cross-linking/mass spectrometry as well as in vitro and cellular functional analyses of the ASCC3-TRIP4 sub-module of ASCC. Unlike the related spliceosomal SNRNP200 RNA helicase, ASCC3 can thread substrates through both helicase cassettes. TRIP4 docks on ASCC3 via a zinc finger domain and stimulates the helicase by positioning an ASC-1 homology domain next to the C-terminal helicase cassette of ASCC3, likely supporting substrate engagement and assisting the DNA exit. TRIP4 binds ASCC3 mutually exclusively with the DNA/RNA dealkylase, ALKBH3, directing ASCC3 for specific processes. Our findings define ASCC3-TRIP4 as a tunable motor module of ASCC that encompasses two cooperating NTPase/helicase units functionally expanded by TRIP4.

Nucleic acid-dependent nucleoside-triphosphatases (NTPases) are pervasively involved in processes related to DNA replication, recombination, genome maintenance, gene expression and co-/post-transcriptional gene regulation[1]. These enzymes exhibit nucleic acid binding, translocating and/or unwinding activities, and are often referred to as DNA or RNA helicases, depending on the nucleic acid specificity[2,3]. In vivo they utilize to these molecular activities to often act as versatile remodelers of nucleic acid–protein complexes[4,5]. The intrinsic molecular mechanisms of nucleic acid-dependent NTPases are diverse, relying on core RecA-like NTPase domains that frequently are functionally expanded by peripheral regions and auxiliary domains, and can be further modulated by interacting regulators[5,6].

[1]Freie Universität Berlin, Institute of Chemistry and Biochemistry, Laboratory of Structural Biochemistry, Takustr. 6, D-14195 Berlin, Germany. [2]Freie Universität Berlin, Institute of Chemistry and Biochemistry, Research Center of Electron Microscopy, Fabeckstr. 36a, D-14195 Berlin, Germany. [3]Universitätsmedizin Göttingen, Department of Molecular Biology, Humboldallee 23, D-37073 Göttingen, Germany. [4]Max-Planck-Institut für Multidisziplinäre Naturwissenschaften, Bioanalytical Mass Spectrometry, Am Fassberg 11, D-37077 Göttingen, Germany. [5]Washington University School of Medicine, Department of Pathology & Immunology and Center for Genome Integrity, 660 S. Euclid Ave, St. Louis, MO 63110, USA. [6]Universitätsmedizin Göttingen, Institut für Klinische Chemie, Bioanalytik, Robert-Koch-Straße 40, D-35075 Göttingen, Germany. [7]Georg-August-Universität, Göttingen Center for Molecular Biosciences, Justus-von-Liebig-Weg 11, D-37077 Göttingen, Germany. [8]Max-Planck-Institut für Multidisziplinäre Naturwissenschaften, Am Fassberg 11, D-37077 Göttingen, Germany. [9]Helmholtz-Zentrum Berlin für Materialien und Energie, Macromolecular Crystallography, Albert-Einstein-Str. 15, D-12489 Berlin, Germany. [10]Present address: Harvard Medical School, Department of Cell Biology, 240 Longwood Avenue, Boston, MA 02115, USA. ✉e-mail: markus.wahl@fu-berlin.de

Several nucleic acid-dependent NTPases are involved in more than one cellular process, affording the potential for functional coordination and cross-regulation[7,8].

Activating signal co-integrator 1 complex (ASCC) subunit 3 (ASCC3) has been characterized as a 3′-to-5′ directional DNA translo-

case/helicase[9,10] that is closely related to the spliceosomal RNA helicase, U5 small nuclear ribonucleoprotein 200 kDa (SNRNP200/BRR2). ASCC3 and SNRNP200 are large enzymes (>2100 residues) that contain a tandem array of Ski2-like helicase cassettes (N-terminal cassette, NC; C-terminal cassette, CC), preceded by ~400-residue N-terminal

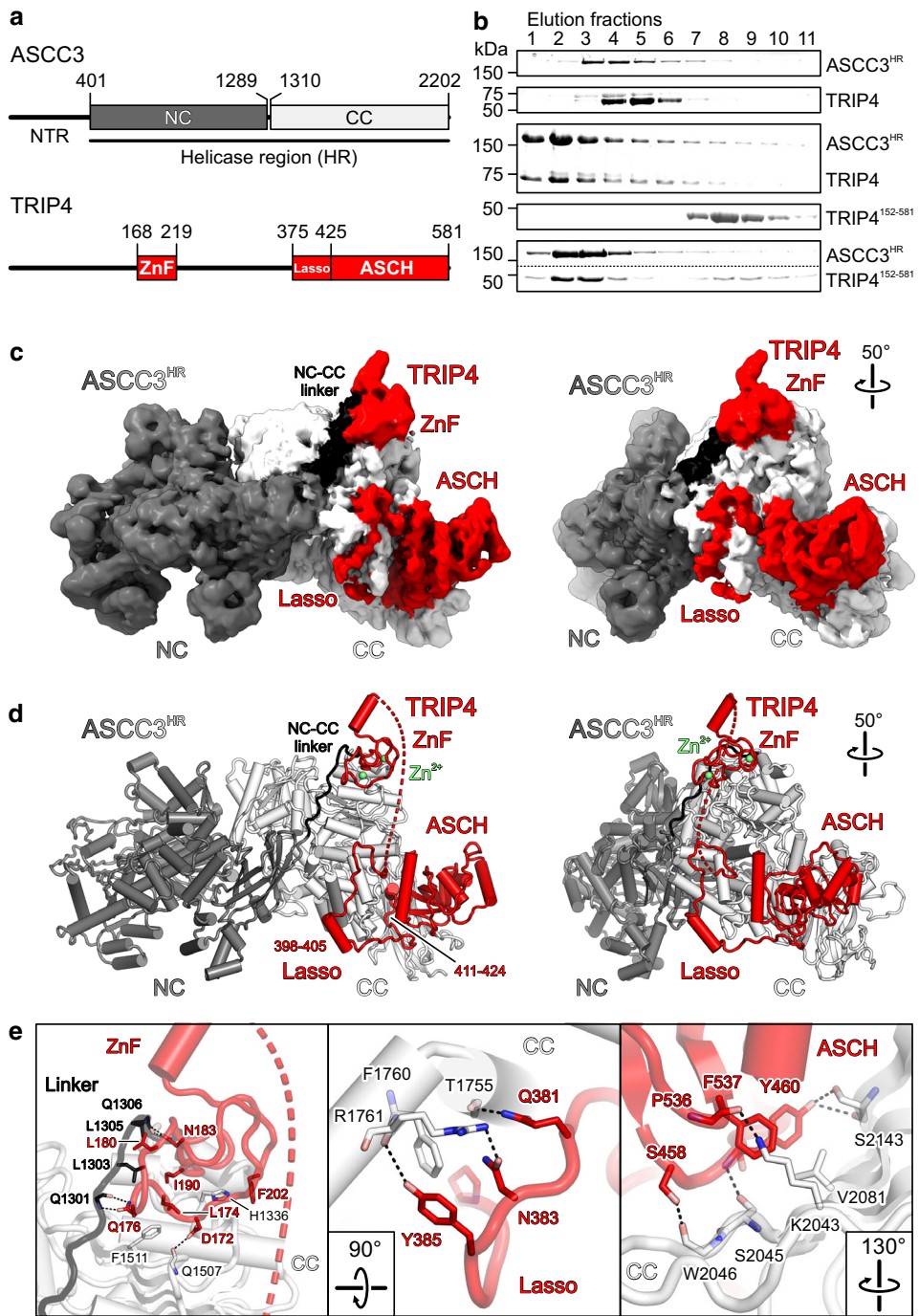

**Fig. 1 | Structure of a ASCC3$^{HR}$–TRIP4 complex. a** Schemes of regions or domains in ASCC3 and TRIP4. Numbers above the schemes, region/domain borders. NTR N-terminal region, NC/CC N-terminal/C-terminal cassettes, HR helicase region, ZnF zinc finger domain, Lasso lasso peptide, ASCH ASC−1 homology domain. **b** SDS-PAGE analyses of analytical SEC elution fractions monitoring the interaction of ASCC3$^{HR}$ with selected TRIP4 variants. Equivalent elution fractions are vertically aligned. Molecular mass markers in kDa are shown on the left; protein bands are identified on the right. In the bottom panel, separate regions of the same gel were spliced together for display purposes (see Source Data file for uncropped gel). Dashed line, splice line. Experiments were repeated independently at least three times with similar results. **c** Overview of the cryoEM reconstruction of the ASCC3$^{HR}$-TRIP4 complex. Regions/domains of ASCC3$^{HR}$ and TRIP4 are labeled. In this and the following panels: ASCC3$^{HR}$ NC, dark gray; ASCC3$^{HR}$ CC, light gray; NC-CC linker, black; TRIP4, red. Rotation symbol, orientation relative to the left panel. **d** Cartoon plot of the ASCC3$^{HR}$-TRIP4 complex model in the same orientations as in (**c**). Zn$^{2+}$ ions, green spheres. **e** Close-up views of the interfaces of the ZnF domain (left), lasso-like peptide (middle) and ASCH domain (right) with ASCC3$^{HR}$. Interacting residues are shown as sticks, colored by atom type, and labeled. Carbon, as the respective protein region; nitrogen, blue; oxygen, light red. Dashed black lines, hydrogen bonds or salt bridges. Rotation symbols, orientations relative to (**c**, **d**), left panels. Source data are provided as a Source Data file.

regions that can auto-inhibit the helicase activities[10–12]. ASCC3 represents a particularly versatile nucleic acid-dependent NTPase in humans that might form functional complexes with overlapping but non-identical sets of interaction partners to support diverse genome maintenance and gene expression processes.

Originally, ASCC3 was discovered as a component of the human ASCC that additionally encompasses subunits ASCC1 (containing RNA-binding KH and RNA ligase-like domains[13]), ASCC2 (containing a K63-linked ubiquitin chain-binding CUE domain[14]) and activating signal co-integrator 1/thyroid receptor-interacting protein 4 (TRIP4)[15]. By associating with basal transcription factors[16], nuclear receptors[16,17] and/or various co-activators[15,16,18,19], ASCC is thought to establish distinct transcription co-activator complexes in response to different cellular conditions[16,18]. Moreover, ASCC3, presumably as part of the ASCC, has been identified as a transcription modulator of antiviral type I interferon-stimulated genes during infections by positive-strand RNA viruses[20], and is involved in the transcriptional response to UV irradiation or to agents that give rise to bulky DNA lesions[21,22].

The ASCC-related ribosome quality control trigger (RQT) complex encompasses ASCC2, ASCC3 and TRIP4, but apparently lacks ASCC1. The RQT complex aids in resolving stalled di-ribosomes or polysomes arising during aberrant translation, where it ultimately splits the stalled lead ribosomes into subunits[23,24]. An analogous RQT complex, comprising RQT2 (ASCC2 ortholog), Slh1p/RQT2 (ASCC3 ortholog), and RQT4 (TRIP4 ortholog) has been identified in yeast[25–28].

In yet another molecular constellation, ASCC3 associates with ASCC1, ASCC2, and the single-stranded (ss) DNA/RNA-specific α-ketoglutarate/iron-dependent dioxygenase, ALKBH3[29,30] to support DNA alkylation damage repair[9,13,31]. Here, ASCC3 generates single-stranded DNA for dealkylation by ALKBH3[9,31]. ASCC3, possibly as part of the same complex, is also required for ALKBH3-dependent removal of $m^1A$ and $m^3C$ modifications from mRNAs[32]. The latter activity of ASCC3 may be linked to its role in ribosome quality control, as ASCC3 has been suggested to help disassemble ribosomes collided on alkylated mRNAs for mRNA dealkylation by ALKBH3[32].

NTPase-fueled remodeling of nucleic acids or nucleic acid–protein complexes by ASCC3 likely constitute central activities for all of the above cellular processes. However, presently the molecular mechanisms underlying ASCC3 nucleic acid helicase activity and its regulation are poorly understood. For example, while in the ASCC3 homolog, SNRNP200, only the NC is an active NTPase/helicase, with the inactive CC acting as an intramolecular helicase co-factor[11,33], both helicase cassettes in ASCC3 may be enzymatically active[9,10,24]. Moreover, while ASCC3 has originally been described as a DNA helicase, it is thought to translocate on mRNA or possibly rRNA during ribosome quality control. However, in a recent structural analysis of yeast RQT–ribosome complexes, RNA binding to either Slh1p helicase cassette remained unresolved[25]. Thus, it is presently unclear whether during ASCC3-related processes the cassettes engage continuous or discontinuous regions of a nucleic acid substrate or perhaps even different nucleic acid molecules, and how their helicase activities may be coordinated. In addition, the precise molecular contexts in which ASCC3 contributes to its various cellular functions, and how these complexes are established is incompletely understood. Finally, it is largely unknown how ASCC3-interacting proteins may influence the ASCC3 helicase activity.

Previously, we showed that ASCC2 associates with a small helical domain in the N-terminal region of ASCC3, an interaction that may auto-regulate ASCC3[10]. Here, we report that the TRIP4 protein binds the ASCC3 helicase region and supports a hitherto unobserved mechanism of nucleic acid translocation/unwinding. Using cryogenic electron microscopy (cryoEM)/single-particle analysis (SPA) and DNA–protein cross-linking/mass spectrometry (CLMS)-based structural analyses as well as systematic protein interaction, DNA binding and unwinding assays, we show that ASCC3 can thread DNA through both of its helicase cassettes. TRIP4 docks to the ASCC3 CC via a zinc finger (ZnF) domain, positioning its ASC-1 homology (ASCH) domain such that it can engage DNA exiting from ASCC3. We also present evidence that TRIP4 and ALKBH3 bind ASCC3 in a mutually exclusive manner and that TRIP4 does not affect ASCC3-dependent DNA alkylation damage repair, demonstrating that ASCC3 indeed assembles different molecular complexes to support different cellular functions.

## Results

### TRIP4 directly associates with the helicase region of ASCC3

Previous interaction mapping suggested that, apart from providing ATPase/helicase activities, ASCC3 may represent a major scaffold for assembling complexes for diverse cellular functions[9,10,13,31]. We therefore tested whether TRIP4, which is implicated with ASCC3 in transcription regulation and ribosome quality control, also directly binds ASCC3 in vitro. While TRIP4 did not stably interact with the ASCC3 N-terminal region (ASCC3^NTR, residues 1–400), it co-eluted with the helicase region of ASCC3 (ASCC3^HR, residues 401-2202) in analytical size-exclusion chromatography (SEC; Fig. 1a, b and Supplementary Fig. 1a, b). We therefore further investigated TRIP4 as a potential direct regulator of the ASCC3 nucleic acid NTPase/helicase activities.

To this end, we reconstituted an ASCC3^HR-TRIP4 complex and determined its atomic structure via cryoEM/SPA at a nominal resolution of 3.4 Å (Fig. 1c, Supplementary Figs. 2 and 3, and Supplementary Table 1). In the cryoEM reconstruction, we could trace residues 401–2183 of ASCC3^HR as well as residues 168–219 and 375–580 of TRIP4 (Fig. 1c, d), capitalizing on AlphaFold-predicted models[34]. ASCC3^HR adopts a structure very similar to the helicase region of SNRNP200 (root mean square deviation [rmsd] of 3.1 Å for 1504 pairs of Cα atoms compared to isolated SNRNP200^HR; PDB ID 4F91; Supplementary Fig. 4)[11]. Like SNRNP200, both ASCC3 helicase cassettes contain consecutive dual RecA-like (RecA1, RecA2), winged-helix (WH), helical bundle (HB), helix−loop−helix (HLH) and immunoglobulin-like (IG) domains and associate to form a compact helicase region (Supplementary Fig. 4). An extended, irregularly structured linker (residues 1290–1309) connects the IG domain of the NC to the RecA1 domain of the CC, running closely along the body of the ASCC3 CC (Fig. 1c, d).

TRIP4 exclusively associates with the CC of ASCC3^HR (Fig. 1c, d). Residues 168–219 of TRIP4 fold into a dual-ZnF domain, with residues C171/C173/H178/C192 and C184/C187/C200/C203 each coordinating a zinc ion (Fig. 1d). The ZnF domain of TRIP4 rests on top of the RecA1 domain of the ASCC3 CC, neighboring the extended linker to the NC and spanning ~757 Å² of interface area, with hydrophobic interactions in the center and hydrophilic interactions at the periphery (Fig. 1e, left). TRIP4 residues 375–424 lack a globular fold and regular secondary structure elements, except for a short helical region in residues 398–405. They form a lasso-like structure around a protruding edge of the C-terminal ASCC3 WH domain, with residues 411–424 inserted deeply into a groove between the RecA1, WH, HB, and IG domains of the ASCC3 CC, spanning ~1914 Å² of interface area with ASCC3^HR (Fig. 1e, middle). TRIP4 residues 411–424 form a support for the C-terminal ASCH domain of TRIP4 (residues 425–578) that further interconnects the C-terminal ASCC3 RecA1, WH, and IG domains, spanning an additional ~1321 Å² of interface area with ASCC3^HR (Fig. 1e, right).

### The ZnF domain is required for stable docking of TRIP4 on ASCC3^HR in vitro

Based on the structure, we designed various TRIP4 fragments to probe the importance of different regions for stable complex formation with ASCC3^HR. Consistent with the cryoEM structure, the N-terminal 80 residues of TRIP4 did not sustain a stable interaction with ASCC3^HR (Supplementary Fig. 1c), while TRIP4 residues 152–581, encompassing the ZnF domain, the lasso-like peptide and the ASCH domain, co-migrated with ASCC3^HR in analytical SEC (Fig. 1b). An N-terminal TRIP4

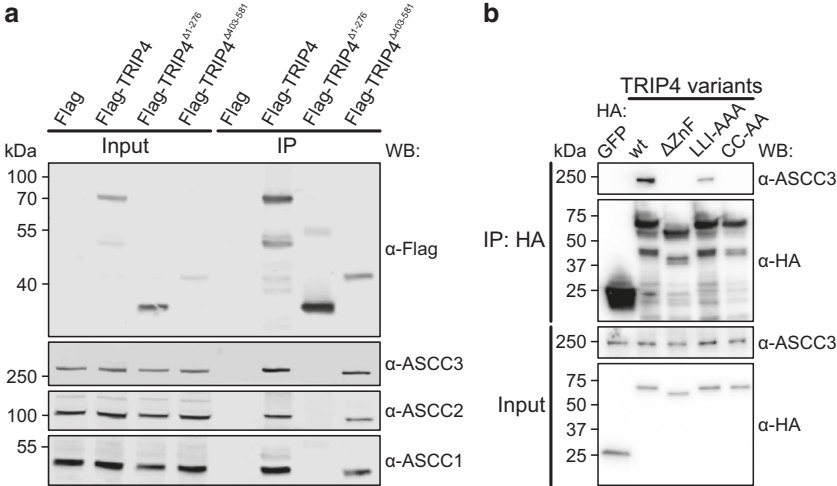

**Fig. 2 | Interactions of TRIP4 variants in cells. a** Western blots (WB) monitoring immunoprecipitation (IP) of ASCC1, ASCC2, and ASCC3 by the indicated N-terminally Flag-tagged TRIP4 variants from the cell extracts. **b** Western blots (WB) monitoring immunoprecipitation (IP) of ASCC3 by the indicated HA-tagged

TRIP4 variants (negative control, GFP). Wt, TRIP4 wild-type; ΔZnF, TRIP4$^{Δ168-219}$; LLI-AAA, TRIP4$^{L174A-L180A-II90A}$; CC-AA, TRIP4$^{C171A-C184A}$. Experiments were repeated independently three times with similar results. Source data are provided as a Source Data file.

region including the ZnF domain (residues 1–230) or the ZnF domain alone (residues 152–230) also stably bound ASCC3$^{HR}$ (Supplementary Fig. 1d). In contrast, C-terminal TRIP4 residues 281–403, 403–581, or 281–581, containing the lasso-like peptide, the ASCH domain or both, did not support stable complex formation with ASCC3$^{HR}$ (Supplementary Fig. 1e), although these regions span a considerably larger interface with ASCC3$^{HR}$ than the ZnF domain (see above). Thus, only the ZnF domain of TRIP4 is required for stable complex formation in vitro, and only upon anchoring via the ZnF domain, the C-terminal ASCH domain and the preceding peptide region of TRIP4 are stably docked on the ASCC3 CC.

## Cellular interaction tests corroborate in vitro interaction patterns

To test the importance of TRIP4 regions for the interaction with ASCC3 and other ASCC subunits in cells, we generated stably transfected Flp-In™ T-REx™ 293 cell lines for the inducible expression of N- or C-terminally Flag-tagged versions of full-length TRIP4 or truncation variants lacking either N-terminal regions including the ZnF domain (TRIP4$^{Δ1-276}$) or lacking the C-terminal ASCH domain (TRIP4$^{Δ403-581}$). Immunofluorescence microscopy showed that all constructs were located to both the cytosol and the nucleus (Supplementary Fig. 5a). We then immunoprecipitated the Flag-tagged TRIP4 variants with α-Flag antibodies and probed the eluates for the presence of other ASCC subunits by western blot. Irrespective of the position of the tag, TRIP4 and TRIP4$^{Δ403-581}$ (lacking the ASCH domain) co-precipitated ASCC1, ASCC2, and ASCC3 (Fig. 2a and Supplementary Fig. 5b). In contrast, no interaction with these ASCC subunits was detected by co-precipitation with TRIP4$^{Δ1-276}$ (lacking the ZnF domain; Fig. 2a and Supplementary Fig. 5b).

To further test the relevance of ASCC3$^{HR}$-TRIP4 contacts observed in our cryoEM structure for the interaction of ASCC3 and TRIP4 in cells, we transfected 293T cells for the expression of N-terminally HA-tagged versions of TRIP4. In these TRIP4 variants, either the ZnF domain was precisely deleted (ΔZnF; deletion of residues 168–219), three residues that engage in hydrophobic interactions with ASCC3$^{HR}$ were exchanged for alanines (LLI-AAA, TRIP4$^{L174A-L180A-II90A}$; Fig. 1e, left) or two cysteines coordinating the first (C171) and second (C184) Zn$^{2+}$ ion were exchanged for alanines (CC-AA, TRIP4$^{C171A-C184A}$). While wild-type (wt) TRIP4 efficiently co-immunoprecipitated endogenous ASCC3, the ΔZnF and CC-AA variants entirely lost the ability to immunoprecipitate

ASCC3, and the ASCC3 interaction of the LLI-AAA variant was strongly reduced (Fig. 2b).

Together, the results of these cellular interaction studies are fully in line with the in vitro ASCC3$^{HR}$–TRIP4 interaction profiles. They confirm that the ZnF domain of TRIP4 is the main ASCC3-interacting domain of TRIP4, via which TRIP4 also seems to be incorporated into larger complexes additionally comprising ASCC1 and/or ASCC2, and suggest that TRIP4, ASCC1 and ASCC2 can concomitantly interact with ASCC3. The observations also confirm that our cryoEM structure closely represents the mode of interaction of ASCC3 and TRIP4 in cells.

## TRIP4 activates ASCC3 helicase activity without influencing ASCC3 ATPase activity

To test the effect of TRIP4 on the helicase activity of ASCC3$^{HR}$, we conducted fluorescence-based unwinding assays in a stopped-flow device using a DNA duplex with a 31-residue 3'-overhang. In the absence of a DNA trap, the observed time traces fit to a double exponential equation, from which we extracted amplitudes ($A_{fast}$ and $A_{slow}$) and rate constants ($k_{fast}$ and $k_{slow}$) for a fast and a slow phases of the reactions, as well as amplitude-weighted unwinding rate constants ($k_{uaw}$)[35–37]. ASCC3$^{HR}$ alone efficiently unwound the substrate DNA ($k_{uaw} = 0.024 \, s^{-1}$), but unwinding was further stimulated 2.3-fold by TRIP4 ($k_{uaw} = 0.054 \, s^{-1}$; Fig. 3a and Supplementary Table 2). TRIP4 increased both the fast and slow rate constants of the unwinding process (Supplementary Table 2). In contrast, both TRIP4$^{1–230}$ (encompassing the ZnF domain), which stably bound ASCC3$^{HR}$ in analytical SEC, as well as TRIP4$^{403–581}$ (encompassing the ASCH domain and preceding peptide), which did not co-migrate with ASCC3$^{HR}$ in analytical SEC, only marginally affected the ASCC3$^{HR}$ helicase activity ($k_{uaw} = 0.030 \, s^{-1}$ and $0.035 \, s^{-1}$, respectively; Fig. 3a and Supplementary Table 2). Thus, while the TRIP4 ZnF domain alone can stably bind to ASCC3$^{HR}$, it does not efficiently activate ASCC3$^{HR}$ helicase activity, for which the lasso-like peptide and ASCH domain are also required.

To test if the biphasic unwinding behavior was due to multiple rounds of unwinding, we repeated the experiments for ASCC3$^{HR}$ and the ASCC3$^{HR}$-TRIP4 complex in the presence of a 50-fold molar excess of a trapping DNA (unlabeled short strand of the duplex). Again, biphasic time traces yielding very similar rate constants as in the absence of a trap were observed (Supplementary Fig. 6a and Supplementary Table 2), suggesting that our initial assays did not monitor DNA re-annealing and re-binding of the helicase machineries. Instead,

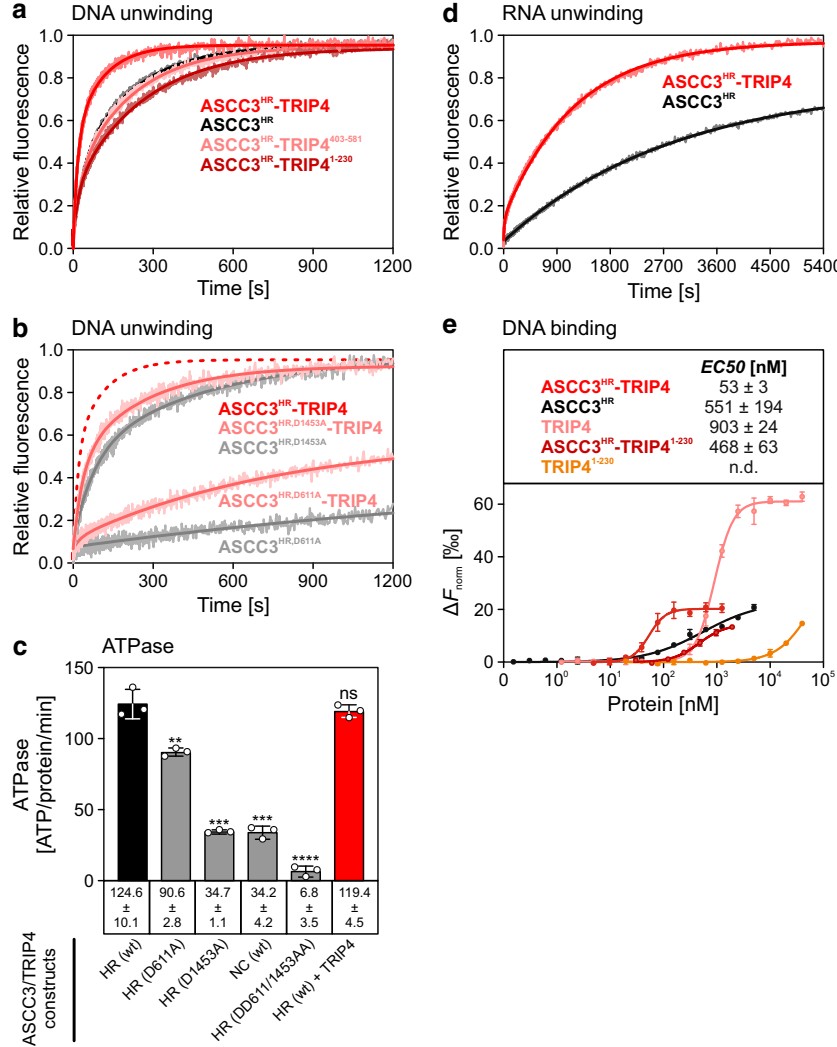

**Fig. 3 | Modulation of ASCC3HR helicase, ATPase, and DNA binding activities by TRIP4. a** Stopped-flow/fluorescence-based DNA unwinding assays (no trap), showing that TRIP4, but not TRIP4[1–230] or TRIP4[403–581], stimulates ASCC3HR DNA helicase activity. In this and analogous experiments in the following, curves show fits of the data to a double exponential equation (fraction unwound = $A_{fast}$*(1 − exp(−$k_{fast}$ * $t$)) + $A_{slow}$ * (1 − exp(−$k_{slow}$ * $t$)); $A_{fast/slow}$, unwinding amplitudes of the fast/slow phases; $k_{fast/slow}$, unwinding rate constants of the fast/slow phases [s⁻¹]; $t$, time [s])[35]. **b** Stopped-flow/fluorescence-based assays (no trap) monitoring DNA unwinding by ASCC3HR constructs, in which either the NC (D611A) or the CC (D1453A) are inactivated, alone or in the presence of TRIP4, showing that both NC and CC exhibit helicase activities that are stimulated by TRIP4. Data for ASCC3[HR,D611A]-based DNA unwinding had been reported previously[10] and are reproduced here to facilitate direct comparison. **c** Apparent DNA-stimulated ATPase rates of ASCC3 constructs alone or in complex with TRIP4 (indicated at the bottom). HR, helicase region; NC, N-terminal cassette. Values represent means (bars) ± SD (lines); individual data points (open circles) for $n = 3$ technical replicates

are shown. Apparent ATPase rates were calculated as described in "Methods" and in Supplementary Fig. 7. Statistical significance was determined by unpaired, two-sided $t$ tests. Significance indicators represent the significance of differences to wt ASCC3HR; **$P = 0.0049$; ***$P = 0.0001$; ****$P < 0.0001$; ns not significant. ASCC3HR constructs, in which either the NC (D611A) or the CC (D1463A) are inactivated, show reduced ATPase activities, and TRIP4 does not significantly enhance the ASCC3HR ATPase. **d** Stopped-flow/fluorescence-based RNA unwinding assays (no trap), showing that ASCC3HR can unwind RNA duplexes and that ASCC3HR-based RNA unwinding is also modulated by TRIP4. **e** MST assays monitoring DNA binding by the indicated ASCC3 and TRIP4 variants or complexes. Data represent means ± SD; $n = 3$ (ASCC3HR), n = 7 (ASCC3HR-TRIP4), $n = 6$ (ASCC3HR-TRIP4[1–230]), $n = 4$ (TRIP4) or $n = 6$ (TRIP4[1–230]) technical replicates. Curves show fits of the data to a Hill model ($F_{norm} = F_{norm,max}$ * $X^h$ / ($EC50^h + X^h$); $F_{norm,max}$, maximum $F_{norm}$ value; $X$, concentration of the protein or protein complex; $h$, Hill slope; $EC50$, concentration needed to achieve a half-maximum binding at equilibrium). Source data are provided as a Source Data file.

slow and fast phases may represent, e.g., an initial productive accommodation of DNA after ATP addition followed by the unwinding process per se, and both phases are stimulated by TRIP4.

Next, we asked which helicase cassette of ASCC3HR is preferentially regulated by TRIP4. To this end, we employed ASCC3HR variants, in which a crucial motif II aspartate of the NC (D611) or CC (D1453) was exchanged for an alanine, abrogating NTPase/helicase activities in the respective cassette[10]. ASCC3[HR,D1453A], bearing an inactive CC, unwound DNA at a reduced rate ($k_{uaw} = 0.011$ s⁻¹), while the unwinding activity of ASCC3[HR,D611A], containing an inactive NC, was strongly reduced ($k_{uaw}$ n.d.; Fig. 3b), suggesting that both cassettes are

required for full ASCC3 helicase activity. Only the construct bearing an inactive CC was stimulated by TRIP4 to quantifiable levels (ASCC3[HR,D1453A]-TRIP4 $k_{unw} = 0.024$ s⁻¹; Fig. 3b and Supplementary Table 2).

NTPase activity associated with both ASCC3HR cassettes was further corroborated by DNA-stimulated ATPase assays. ASCC3[HR,D1453A] (inactive CC) and ASCC3[HR,D611A] (inactive NC) exhibited ~28 and ~73% of the DNA-stimulated ATPase activity of wt ASCC3HR, while the DNA-stimulated ATPase activity of the ASCC3[HR,DD611/1453AA] variant, with motif II changes in both cassettes, was negligible (Fig. 3c and Supplementary Fig. 7). As expected if the implemented residue exchanges selectively

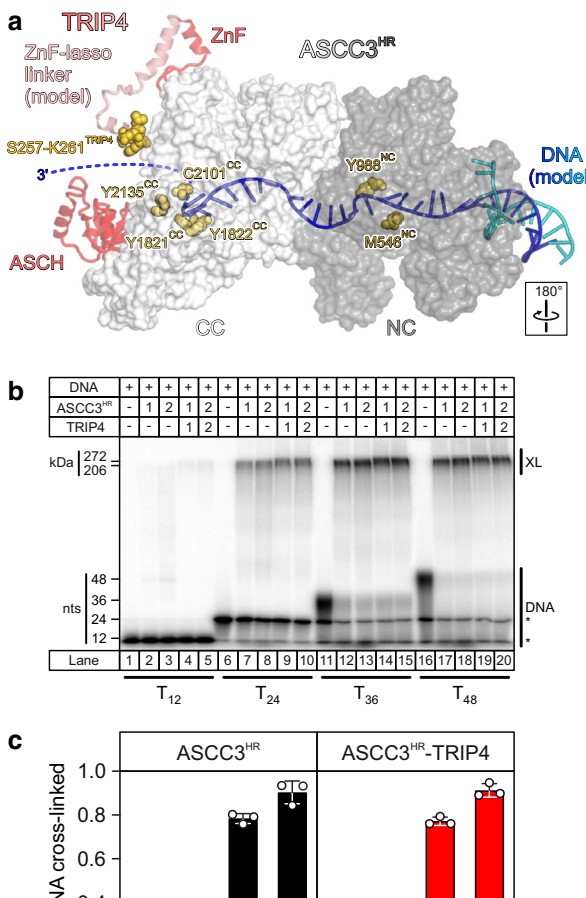

**Fig. 4 | Path of DNA through the ASCC3[HR]–TRIP4 complex. a** Model of a ASCC3[HR]–TRIP4–DNA complex. ASCC3[HR], semi-transparent surface view (NC, dark gray; CC, light gray); TRIP4, semi-transparent cartoon view (red) with part of the TRIP4 ZnF-lasso linker region (light red) modeled with AlphaFold;[34] DNA modeled according to the DNA-bound Hel308 DNA helicase (PDB ID 2P6R),[40] cartoon (blue and cyan); UV-cross-linked residues as identified by MS, spheres (gold). Cross-linked residues line the modeled path of the ssDNA region through both cassettes and exiting the CC near the TRIP4 ASCH domain (dashed blue line). Rotation symbol, orientation relative to Fig. 1c, d, left panels. **b** SDS-PAGE analysis monitoring UV-induced cross-linking of radio-labeled oligo-T DNAs (indicated at the bottom) to ASCC3[HR] (lanes 2, 3, 7, 8, 12, 13, 17, 18) or to the ASCC3[HR]–TRIP4 complex (lanes 4, 5, 9, 10, 14, 15, 19, 20). Lanes 1, 6, 11, 16, DNAs alone. Numbers above the gel indicate the amounts of ASCC3[HR] and TRIP4 (1, 100 nM; 2, 200 nM) added to 4.3 nM radio-labeled DNA. Molecular mass markers in kDa or nucleotides (nts) are shown on the left, labeled bands are identified on the right. Asterisks, bands in T24, T36 and T48 samples representing truncated synthetic products and DNA degraded during labeling and/or UV irradiation. **c** Quantification of the data in (**b**) obtained with samples containing 200 nM ASCC3[HR] or ASCC3[HR]–TRIP4. Bars represent means ± SD; $n$ = 3 technical replicates. Individual data points are shown as open circles. Source data are provided as a Source Data file.

abrogated ATPase activity in the respective cassette, the ATPase activity of ASCC3[HR,D1453A] (inactive CC) closely matched the ATPase activity of the isolated wt NC (Fig. 3c and Supplementary Fig. 7). As we failed to produce the ASCC3 CC in isolation, a similar comparison could not be drawn between ASCC3[HR,D611A] (inactive NC) and isolated wt

CC. Irrespectively, in contrast to the helicase activity, the stimulated ATPase activity of ASCC3[HR] was not further enhanced by TRIP4. Thus, TRIP4 activates ASCC3[HR] helicase activity without affecting its ATPase activity.

## ASCC3 exhibits TRIP4-modulated RNA unwinding activity

As ASCC3 is thought to translocate RNA during ribosome quality control, we also tested whether ASCC3[HR] can unwind RNA duplexes and whether this activity is likewise modulated by TRIP4. Using an RNA duplex of analogous sequence as the employed DNA substrate, stopped-flow/fluorescence-based unwinding assays in the absence of a trap revealed that ASCC3 indeed also exhibits TRIP4-modulated RNA unwinding activity (Fig. 3d). Again, biphasic time traces were observed, but the influence of TRIP4 on ASCC3[HR]-mediated RNA unwinding was complex. In the presence of TRIP4, the rate constant for the fast phase was decreased while the rate constant for the slow phase was enhanced; at the same time, TRIP4 led to an approximately tenfold increased amplitude for the fast component (Supplementary Table 2). Thus, while the amplitude-weighted RNA unwinding rate constant of ASCC3[HR] ($k_{uaw}$ = 0.073 s$^{-1}$) was reduced in the presence of TRIP4 ($k_{uaw}$ = 0.046 s$^{-1}$), a larger fraction of ASCC3[HR] molecules productively engaged the RNA substrate in the presence of TRIP4, leading to a larger fraction of RNA duplexes being unwound.

## TRIP4 enhances DNA engagement by ASCC3

The TRIP4 ASCH domain constitutes a putative nucleic acid-binding domain[38,39] that might contribute to DNA binding by an ASCC3–TRIP4 complex. To test this notion, we conducted comparative DNA binding assays using a Cy5-labeled T48 DNA oligomer in microscale thermophoresis (MST) assays. Both ASCC3[HR] and TRIP4 individually bound T48 DNA with an $EC50$ of 551 ± 194 nM and 903 ± 24 nM, respectively (Fig. 3e). The T48 DNA affinity of the ASCC3[HR]-TRIP4 complex ($EC50$ = 53 ± 3 nM) was approximately tenfold stronger compared to isolated ASCC3[HR]. In contrast, TRIP4[1–230] (containing the ZnF domain but lacking the lasso-like peptide and ASCH domain) exhibited very low DNA affinity ($EC50$ n.d.) and did not significantly influence the DNA affinity of ASCC3[HR] ($EC50$ = 468 ± 63 nM). Therefore, additional nucleic acid contacts established via the TRIP4 ASCH domain could modulate the ASCC3[HR]-mediated DNA unwinding mechanism and/or might influence initial substrate engagement by ASCC3[HR].

## DNA can be threaded through both ASCC3 helicase cassettes and along TRIP4

We failed to obtain cryoEM structures of ASCC3[HR] or ASCC3[HR]-TRIP4 in complex with ssDNA or with double-stranded (ds) DNA bearing a 3′-ss overhang. We thus modeled a putative path of ssDNA through ASCC3[HR] by superimposing a structure of the Hel308 DNA helicase in complex with DNA (PDB ID 2P6R)[40] on both ASCC3[HR] cassettes, transferring the Hel308 DNA substrates to both cassettes of the ASCC3[HR] model and removing the dsDNA portion from the DNA transferred to the CC (Fig. 4a). In the resulting model, the ssDNA region transferred to the NC could be pseudo-continuous with the ssDNA region transferred to the CC, indicating that a longer ssDNA could be threaded consecutively through both helicase cassettes and might exit the CC close to the TRIP4 ASCH domain (Fig. 4a). The model suggested that a minimum of about 24 nucleotides of ssDNA are required to traverse the two cassettes and TRIP4. In contrast, lateral entry of ssDNA to the CC, circumventing the NC, is blocked in the conformation of ASCC3[HR] observed in our cryoEM structure. A requirement for DNA to enter the CC via the preceding NC would be consistent with the larger effect on helicase activity we observed upon inactivating the NC alone as compared to a ASCC3[HR] variant containing only an inactive CC (Fig. 3b).

To test if, during unwinding, ASCC3[HR] and ASCC3[HR]–TRIP4 might thread single-stranded DNA through both helicase cassettes, and in the

latter case along the TRIP4 ASCH domain, we conducted ultraviolet (UV) radiation-induced cross-linking of $ASCC3^{HR}$ and $ASCC3^{HR}$-TRIP4 to variable-length, single-stranded oligo-T DNAs ($T_{12}$, $T_{24}$, $T_{36}$, $T_{48}$; Fig. 4b). Both $ASCC3^{HR}$ and $ASCC3^{HR}$-TRIP4 did not efficiently cross-link to $T_{12}$ ssDNA and showed stepwise increased cross-linking to $T_{24}$, $T_{36}$, and $T_{48}$ DNAs (cross-link efficiencies of ~30%, 80%, and 90%, respectively; Fig. 4b, c). TRIP4 alone did not efficiently cross-link to any of the DNA samples. These observations are consistent with the notion that a ssDNA region sufficiently long to traverse both cassettes is required for DNA to be efficiently engaged by $ASCC3^{HR}$ or $ASCC3^{HR}$-TRIP4.

Next, we subjected $ASCC3^{HR}$ or $ASCC3^{HR}$-TRIP4 after UV-induced cross-linking to $T_{48}$ ssDNA to DNase/protease digestion, followed by mass spectrometric analysis of cross-linked peptide-DNA conjugates. We observed one cross-linked peptide each in TRIP4 (region connecting ZnF and lasso), the RecA1 domain of the $ASCC3^{HR}$ NC (corresponding to helicase motif Ia), the N-terminal WH domain and the C-terminal WH domain, as well as two cross-linked peptides in the CC IG domain (Table 1). With the exception of the TRIP4 peptide, we could identify one or two specific cross-linked residues in these peptides ($RecA1^{NC}$, M546; $WH^{NC}$, Y988; $WH^{CC}$, Y1821 and Y1822; $IG^{CC}$, C2101, and Y2135; Table 1). The cross-linked residues and the modeled cross-linked TRIP4 peptide are positioned closely along the path of the modeled DNA (Fig. 4a). Together, these observations are consistent with the idea that during unwinding, ssDNA is threaded through both helicase cassettes and along TRIP4 in the vicinity of the ASCH domain. It is, however, also possible that $ASCC3^{HR}$ may undergo conformational changes upon binding to ssDNA of sufficient length, so that the substrate can engage the NC and CC independently. Irrespectively, direct TRIP4-DNA contacts during ASCC3–TRIP4-mediated DNA unwinding are a possible source of TRIP4's direct influence on the unwinding process.

## TRIP4 and ALKBH3 support ASCC core subunits in distinct cellular functions

Present data suggest that ASCC core subunits may associate with different auxiliary proteins to participate in distinct genome maintenance and gene expression processes. More specifically, TRIP4 has so far been found associated with ASCC3-dependent transcription regulation[15,16,18,19] and ribosome quality control[23,24,26], while ALKBH3 is associated with ASCC3 during DNA dealkylation repair[9,31]. We therefore wondered whether TRIP4 and ALKBH3 might bind ASCC3 in a mutually exclusive manner. To test this notion, we conducted competitive SEC-based interaction studies. TRIP4 and ALKBH3 did not co-migrate during SEC (Fig. 5a). A portion of ALKBH3 stably associated with $ASCC3^{HR}$ in SEC, but failed to be incorporated into a pre-formed

$ASCC3^{HR}$–TRIP4 complex (Fig. 5a). These findings suggest that TRIP4

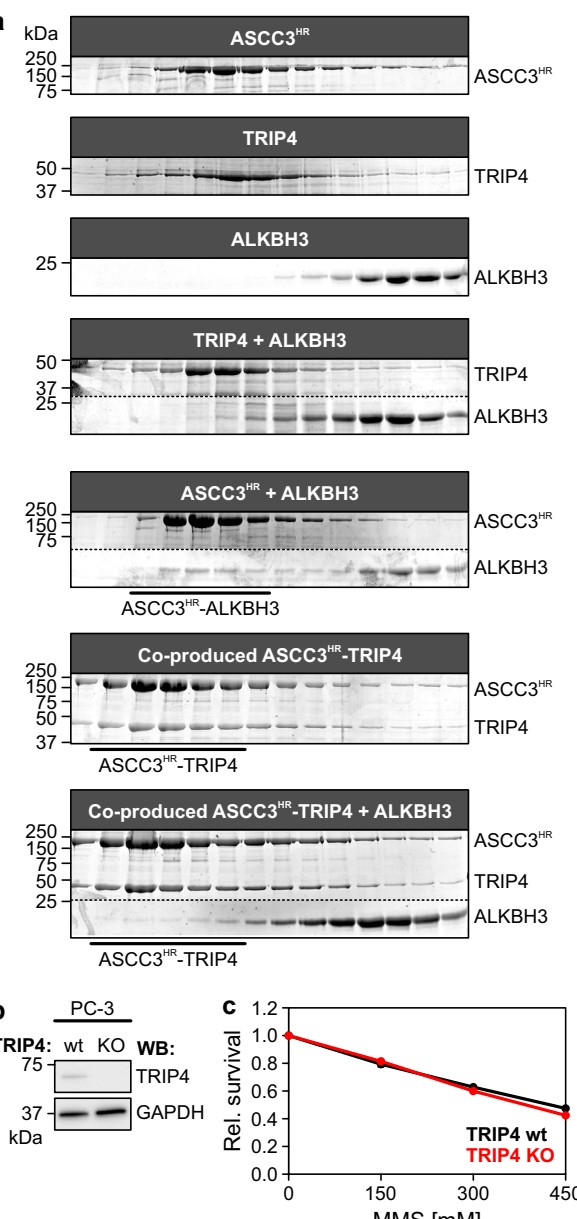

**Fig. 5 | TRIP4 and ALKBH3 may conscribe ASCC core subunits for distinct cellular functions. a** SDS-PAGE analyses of analytical SEC elution fractions monitoring the competitive binding of TRIP4 and AlkBH3 to $ASCC3^{HR}$. Throughout all panels, equivalent elution fractions are vertically aligned. Input samples are identified on top of each run. Molecular mass markers in kDa are shown on the left; protein bands are identified on the right. Stable complexes eluting from some analytical SEC runs are identified below the respective gels. For some analytical SEC runs, separate regions of the same gel were spliced together for display purposes (see Source Data file for uncropped gels). Dashed lines, splice lines. TRIP4 and AlkBH3 do not stably interact (run 4). AlkBH3 and TRIP4 form stable binary complexes with $ASCC3^{HR}$ (runs 5 and 6). AlkBH3 is excluded from a pre-formed $ASCC3^{HR}$-TRIP4 complex (run 7). Experiments were repeated independently at least three times with similar results. **b** Western blots documenting CRISPR/Cas9-mediated KO of TRIP4. GAPDH was used as a loading control. **c** Assay comparing the relative degree of viability of TRIP4 wt and KO PC-3 cells in the presence of increasing concentrations of MMS. TRIP4 wt cells, black; TRIP4 KO cells, red. Values represent means ± SD; $n$ = 5 technical replicates. Error bars are hidden by data points. Source data are provided as a Source Data file.

## Table 1 | DNA–protein cross-links identified by MS

| Cross-linked pepide[a] | Cross-linked residue | Trial | Domain or region | Motif[b] |
|---|---|---|---|---|
| **TRIP4** | | | | |
| 257-SGLEK-261 | n.i. | 1 | ZnF-lasso linker | – |
| **ASCC3**$^{HR}$ | | | | |
| 541-ALAAE**M**TDYFSR-552 | M546 | 2 | $RecA1^{NC}$ | Ia |
| 984-TASH**Y**YIK-991 | Y988 | 1, 2 | $WH^{NC}$ | – |
| 1818-IAS**Y**YYLK–1825 | Y1821 | 2 | $WH^{CC}$ | – |
| 1818-IASY**Y**YLK–1825 | Y1822 | 2 | $WH^{CC}$ | – |
| 2096-GKPES**C**AVTPR-2106 | C2101 | 2 | $IG^{CC}$ | – |
| 2133-VG**Y**IR-2137 | Y2135 | 1, 2 | $IG^{CC}$ | – |

*n.i.* not identified.

[a]Cross-linked residues, bold and underlined.

[b]NC helicase motif Ia, residues 536–546.

and ALKBH3 engage ASCC3[HR] in a mutually exclusive manner, possibly by taking advantage of overlapping binding sites, and that TRIP4 might associate more strongly with ASCC3[HR] than ALKBH3.

To test if ALKBH3 also modulates ASCC3-mediated DNA unwinding, we repeated stopped-flow/fluorescence-based DNA unwinding assays with ASCC3[HR] in the presence of ALKBH3 and in the presence of a DNA trap. ALKBH3 did not significantly alter the amplitudes or rates of the fast and slow phases nor the amplitude-weighted unwinding rate constant of the biphasic unwinding reaction (Supplementary Fig. 6b and Supplementary Table 2).

To further test the idea that either TRIP4 or ALKBH3 associates with ASCC core subunits depending on the particular ASCC-dependent cellular process, we explored the effect of TRIP4 on DNA dealkylation damage repair, where ALKBH3 is known to be involved. To this end, we knocked out TRIP4 via CRISPR/Cas9-based genome engineering in human PC-3 cells (Fig. 5b) and tested the response of the edited and parental cells to methyl methanesulfonate (MMS) that elicits DNA alkylation damage by methylating deoxyguanine (N7) and deoxyadenine (N3). A TRIP4 knockout (KO) did not impact cell survival in the presence of even high concentrations of MMS (Fig. 5c), suggesting that TRIP4 may not be involved in ASCC3/ALKBH3-mediated DNA dealkylation[9]. Together, these observations suggest that TRIP4 and ALKBH3 represent mutually exclusive, process-specific ASCC3 interactors that direct ASCC3 helicase activity toward transcriptional events and ribosome rescue or towards DNA dealkylation damage repair, respectively, but that only TRIP4 modulates ASCC3 helicase activity.

## Discussion

The large nucleic acid-dependent NTPase, ASCC3, exhibits striking homology to the spliceosomal RNA helicase, SNRNP200, and the two proteins represent the only known human members of a unique subfamily of Ski2-like helicases that possess tandem helicase cassettes. Here, we showed by cryoEM-based structural analysis that ASCC3 indeed contains a dual-cassette helicase region that closely resembles the analogous region of SNRNP200, at least in the absence of factors other than TRIP4. In line with previous observations[9,10], our systematic ATPase and DNA unwinding assays strongly suggest that, in contrast to SNRNP200, both ASCC3 cassettes are active ATPases and helicases. Supporting this notion, previous studies suggested a role for ATPase activities in one or both cassettes of ASCC3 or its yeast ortholog, Slh1p, during ribosome quality control. E.g., reduced ribosome association was found for an ASCC3 variant with an ATPase-deficient NC[23], and selective abrogation of the ATPase activity in the ASCC3 NC or CC led to reduced disruption of stalled ribosomes[24]. Likewise, the ATPase activity of the Slh1p NC was shown to be required for ribosome dissociation[26,27,41] or completion of subunit splitting[25].

Our DNA–protein CLMS analyses are consistent with a model in which ASCC3 translocates ssDNA during DNA unwinding, threading one DNA strand consecutively through both helicase units. In principle, our data would also be consistent with the two helicase cassettes unwinding DNA independently of each other. However, in the ASCC3[HR] conformation observed here, direct accommodation of ssDNA at the CC is blocked by the NC. Thus, for the latter scenario, ASCC3 would have to undergo a large conformational rearrangement that leads to a separation of its helicase cassettes if ssDNA were to be captured by the CC without being first threaded through the NC. As ASCC3 interacts with different proteins and substrate complexes in different functional contexts, which could provoke conformational changes in ASCC3, it is conceivable that in certain scenarios the helicase activity of either individual cassette is employed, while in others the two helicase cassettes operate in tandem. Furthermore, in a given functional scenario the two cassettes may even translocate the same or different nucleic acid molecules (see also below).

The CC of SNRNP200 serves as an interaction platform for numerous proteins, several of which inhibit its NC helicase activity from a distance[42–44]. In contrast, the C-terminal Jab1 domain of the large spliceosomal PRPF8 scaffold that can activate the SNRNP200 helicase directly binds the active NC[45,46]. Here, we find that similar to the situation in SNRNP200, the ASCC3 CC serves as a binding platform for the TRIP4 protein. TRIP4 predominantly latches onto ASCC3 via its ZnF domain, allowing the positioning of an ASCH domain close to the presumed DNA exit of the ASCC3 CC with the help of the intervening lasso peptide. However, unlike many proteins that bind the SNRNP200 CC, we show that TRIP4 stimulates ASCC3 helicase activity. The ZnF docking domain is insufficient for helicase stimulation, which also requires C-terminal TRIP4 regions including the ASCH domain.

Our functional analyses suggest that TRIP4 could modulate the ASCC3 helicase by multiple, not mutually exclusive mechanisms, that seem to depend on nucleic acid binding by the ASCH domain[38,39]. First, the DNA affinity of ASCC3[HR] is enhanced in the presence of TRIP4 due to the ASCH domain. TRIP4 could, thus, support ASCC3 loading onto substrate DNA. Second, our DNA–protein CLMS data support the notion that the ASCH domain or neighboring regions may facilitate DNA exit from the ASCC3 CC, whereby TRIP4 might influence the ASCC3-mediated unwinding process, as observed.

Cooperation between both helicase cassettes and activation of ASCC3 helicase activity by TRIP4 may be required to unfold sufficiently strong or appropriately coordinated motor activity during transcription regulatory processes and ribosome quality control, where both ASCC3 and TRIP4 are involved. While the targets of ASCC3's motor activity during transcriptional regulation are presently unknown, during ribosome quality control, ASCC3's ATP-dependent motor activity is essential for the disassembly of the lead ribosome in collided di-somes or polysomes[23,24,26]. As no DNA is involved in this process, ASCC3 most likely translocates mRNA or rRNA regions. Indeed, we demonstrate that ASCC3 also exhibits TRIP4-modulated RNA unwinding activity in vitro, albeit reduced compared to its DNA helicase activity. Furthermore, our analyses show that inactivation of either ASCC3 cassette leads to partial loss of ASCC3 helicase activity. Thus, splitting of ribosomes by translocating mRNA or rRNA may require (a) ASCC3 resorting to a translocation mode that involves both active cassettes on the same or on different RNA molecules, (b) additional stimulation by TRIP4 and/or (c) stimulation by another factor that promotes ASCC3 RNA translocation.

Recent cryoEM structures of yeast RQT–ribosome complexes revealed that prior to ribosome splitting the yeast ASCC3 ortholog, Slh1p, can adopt a more open conformation with fewer direct interactions between the two helicase cassettes as observed in our human ASCC3–TRIP4 complex structure[25]. While in the imaged conformations both Slh1p helicase cassettes are potentially accessible to an RNA substrate, no corresponding substrate density was observed at either Slh1p cassette[25]. In the observed conformations, mRNA could apparently be accommodated directly at the Slh1p CC, but an Slh1p variant harboring an ATPase-deficient NC (Slh1p[K361R]) was required to capture RQT–ribosome complexes at a stage preceding ribosome splitting[25], indicating that the NC ATPase/helicase activity is also required for the splitting reaction. Thus, whether both cassettes or only one of them translocate mRNA or whether one cassette engages mRNA while the other operates on an rRNA region during ribosome splitting remains to be elucidated.

Findings reported here also underscore the notion that ASCC3 engages in mutually exclusive interactions with different partners to participate in different processes. We find that TRIP4, which collaborates with ASCC3 during transcriptional regulation and ribosome quality control, binds to ASCC3 in a manner that is mutually exclusive to ALKBH3, which capitalizes on the ASCC3 helicase activity during DNA alkylation damage repair. Unlike TRIP4, ALKBH3 does not modulate the ASCC3 helicase in vitro. Consistent with the idea of these two factors associating with ASCC3 in different functional scenarios, we also show that TRIP4 does not impact cell sensitivity to an alkylating

agent, unlike ALKBH3 or ASCC3[9,13]. As TRIP4 seems to associate more stably with ASCC3[HR] than ALKBH3, it remains to be seen if additional factors may aid ALKBH3 in displacing TRIP4 for DNA dealkylation damage repair. Additional interactors may favor a conformation of ASCC3 that exhibits altered ALKBH3 affinity. It is also possible that the protein interactions of ASCC3 may be dynamically regulated by post-translational modifications or by the recruitment of subsets of factors to specific sub-cellular compartments. Both of the latter principles have been shown to play a role during ASCC3-related cellular processes[13,14,26,31,32,41,47].

# Methods

## Molecular cloning

DNA fragments encoding ASCC3[HR] (wt, D611A, D1453A or D611A-D1453A) or ASCC3[NC] were cloned into a pFL vector for expression as N-terminally His$_{10}$-tagged, TEV-cleavable proteins via recombinant baculoviruses in insect cells[10]. A DNA fragment encoding full-length TRIP4 was PCR-amplified from a synthetic gene (IDT; Supplementary Table 3) and inserted into the pETM-11 or pIDS vectors (EMBL, Heidelberg) for expression as an N-terminally His$_6$-tagged, TEV-cleavable protein. See Supplementary Table 4 for PCR primers used. The pIDS-*trip4* construct was Cre-recombined with pFL-*ascc3*[HR] for co-expression via a recombinant baculovirus in insect cells. DNA fragments encoding TRIP4[1–80], TRIP4[1–230], TRIP4[152–230], TRIP4[152–581], TRIP4[281–403], TRIP4[281–581] or TRIP4[403–581] were amplified via PCR from the pETM-11-*trip4*, and re-cloned into the pETM-11 vector. A DNA fragment encoding full-length ALKBH3 was PCR-amplified from a cDNA library of human HeLa cells and inserted into the pETM-11 vector for expression as an N-terminally His$_6$-tagged, TEV-cleavable protein. All constructs were verified by Sanger sequencing.

For the preparation of a *trip4* sgRNA vector, we followed a previously established method[48], cloning the target sequence into the pLenti-CRISPRV2 vector[49]. Primers used for generating the DNA fragment containing the target sequence is shown in Supplementary Table 4.

For expression of HA-tagged TRIP4 variants, DNA fragments encoding wt or ΔZnF TRIP4 were cloned into pENTR-3C using a synthetic gene (IDT; Supplementary Table 3). Vectors encoding HA-tagged variants TRIP4[L174A-L180A-II90A] or TRIP4[C171A-C184A] were created using the In-Fusion Snap Assembly mutagenesis kit (Takara Bio 683945). Each construct was then cloned into pHAGE-HA-Blast vector[31] via Gateway recombination. All constructs were verified by Sanger sequencing.

## Generation of cell lines

Stably transfected Flp-In™ T-REx™ 293 cell lines (Thermo Fisher Scientific R78007) for the tetracycline-inducible expression of TRIP4 variants with N-terminal 2xFlag-His$_6$ or C-terminal His$_6$-2xFlag tags were generated according to the manufacturer's guidelines[10]. Transfection of the parental cell line was done using X-tremeGENE HP DNA Transfection Reagent (Sigma-Aldrich). After hygromycin-based selection of cells that had genomically integrated the expression cassette, tetracycline-induced expression of the tagged proteins was confirmed by western blotting using a monoclonal α-Flag-M2 antibody (Sigma-Aldrich F3165; 1:7500). For expression of HA-tagged TRIP4 variants, the pHAGE-HA-*trip4* vectors encoding HA-tagged TRIP4[wt], TRIP4[ΔZnF], TRIP4[L174A-L180A-II90A] or TRIP4[C171A-C184A], were transfected into 293T cells (ATCC CRL-3216) using Transit293 transfection reagent (Mirus Bio).

## CRISPR/Cas9-based genome editing

The *trip4* sgRNA expression vector was transfected into the Lenti-X 293T cell line (Takara Bio 632180) together with psPAX2 and pCMV-VSVG (Addgene) for lentivirus production. The virus-containing culture medium was collected 72 h post-transfection. Human PC-3 cells (ATCC CRL-1435) were infected with the viral medium and individual

clones were selected in 96-well plates. The single KO colonies were analyzed by western blot using an α-TRIP4 antibody (Santa Cruz sc-376916; 1:1000).

## Recombinant protein production and purification

ASCC3[HR] variants and ASCC3[NC] were produced in High Five cells (Thermo Fisher Scientific B85502) via recombinant baculoviruses produced in Sf9 cells (Thermo Fisher Scientific 11496015)[10]. Cell pellets were re-suspended in 20 mM HEPES-NaOH, pH 7.5, 500 mM NaCl, 10 mM imidazole, 1 mM DTT, 8.6% (v/v) glycerol (lysis buffer 1), supplemented with cOmplete™ protease inhibitors (Roche) and lysed by sonication using a Sonopuls Ultrasonic Homogenizer HD (Bandelin). The lysate was cleared by centrifugation and filtration. The protein of interest (POI) was captured on Ni$^{2+}$-NTA resin in a gravity flow column, washed with lysis buffer 1 and eluted with lysis buffer 1 containing 400 mM imidazole. Fractions enriched for the POI were supplemented with 1/10 (w/w) TEV protease and dialyzed against 20 mM HEPES-NaOH, pH 7.5, 500 mM NaCl, 1 mM DTT, 8.6% (v/v) glycerol (dialysis buffer) overnight. The sample was then diluted to 100 mM NaCl and loaded onto a HiTrap Heparin HP column (Cytiva), pre-equilibrated with lysis buffer 1 containing 100 mM NaCl. After washing with lysis buffer 1 containing 100 mM NaCl, the POI was eluted with a linear gradient to lysis buffer 1 containing 1.5 M NaCl. The fractions containing the POI were pooled and concentrated with a centrifugal concentrator (100 kDa molecular mass cutoff). The concentrated sample was further purified by SEC on a Superdex 200 10/300 GL column (Cytiva) in 20 mM HEPES-NaOH, pH 7.5, 250 mM NaCl, 5% (v/v) glycerol, 1 mM DTT (SEC buffer). Fractions containing the POI were combined, concentrated, aliquoted, flash-frozen in liquid nitrogen, and stored at −80 °C.

For the preparation of the ASCC3[HR]-TRIP4 complex, TRIP4 was co-produced with ASCC3[HR] in High Five cells. Cell pellets were re-suspended in lysis buffer 1 supplemented with cOmplete™ protease inhibitors, 1 mM DTT and 20 mM imidazole. The samples were lysed by sonication, then the suspension was centrifuged at 56,000×*g* for 1 h, the soluble extract was further filtered through 0.8 μM pore size membrane filters (Millipore). The filtered fractions were collected and incubated with Ni$^{2+}$-NTA resin pre-equilibrated with lysis buffer 1 for 2 h with gentle rotation at 4 °C. POI-bound resin was loaded on a gravity flow column, washed with lysis buffer 1 and the POI was eluted with lysis buffer 1 containing 400 mM imidazole. To remove the His$_{6/10}$-tags, 1/10 (w/w) of TEV protease was added and the sample and dialyzed against dialysis buffer overnight. Subsequently, the sample was diluted to 50 mM NaCl and loaded on a 5 ml StrepTrap HP column (Cytiva) pre-equilibrated with lysis buffer 1 containing 50 mM NaCl. After washing with lysis buffer 1 containing 50 mM NaCl, the POI was eluted in a linear gradient to lysis buffer 1 containing 1.5 M NaCl. Fractions containing the POI were combined, diluted to 50 mM NaCl, loaded on a 5 ml HiTrap Heparin HP column, washed and eluted in a linear gradient with lysis buffer 1 containing 1.5 M NaCl. Fractions containing the POI were pooled, concentrated and further purified by SEC on a Superdex 200 10/600 GL column (Cytiva) in 20 mM HEPES-NaOH, pH 7.5, 300 mM NaCl, 1 mM DTT. Fractions containing the POI were combined, concentrated, aliquoted, flash-frozen in liquid nitrogen and stored at −80 °C.

For the production of isolated TRIP4 variants, the corresponding pETM-11 vectors were transformed into *Escherichia coli* BL21 (DE3) cells by electroporation for protein production via auto-induction at 18 °C[50]. Cells were harvested when cultures reached an optical density (600 nm) of 10. Cell pellets were re-suspended in lysis buffer 1 and supplemented with cOmplete™ protease inhibitors. After sonication, the lysate was cleared by centrifugation. The POI was captured on Ni$^{2+}$-NTA resin in a gravity flow column, washed with lysis buffer 1 and eluted with lysis buffer 1 containing 400 mM imidazole. Fractions enriched for the POI were supplemented with 1/10 (w/w) TEV protease

and dialyzed against dialysis buffer overnight. Dialyzed samples were passed through a Ni$^{2+}$-NTA gravity flow column to remove the cleaved His$_6$-tag and TEV. For TRIP4, TRIP4$^{152-581}$, TRIP4$^{281-581}$, and TRIP4$^{403-581}$ fragments, the samples were diluted to 100 mM NaCl, loaded on a HiTrap Heparin HP column, washed and eluted in a linear gradient to lysis buffer 1 containing 1.5 M NaCl. Fractions containing the POI were combined, concentrated and further purified on a Superdex 200 16/600 GL column in SEC buffer.

For purification of the TRIP4$^{1-80}$, TRIP4$^{1-230}$, TRIP4$^{152-230}$, and TRIP4$^{281-403}$ fragments, the Heparin column step was omitted and the final gel filtration was conducted in 20 mM HEPES-NaOH, pH 7.5, 150 mM NaCl, 1 mM DTT on a HiLoad 16/60 Superdex 75 pg column (Cytiva).

For the production of ALKBH3, the corresponding pETM-11 vector was transformed into *E. coli* C2566 cells by electroporation for protein production via IPTG induction at 37 °C. Cell pellets were re-suspended in 20 mM TRIS-HCl, pH 7.5, 500 mM NaCl, 10 mM imidazole, 1 mM DTT, 0.1 mM PMSF (lysis buffer 2), and lysed by sonication. The lysate was cleared by centrifugation. The supernatant was loaded onto a Ni$^{2+}$-NTA column, washed with lysis buffer 2, and the POI was eluted with a linear gradient to lysis buffer 2 containing 400 mM imidazole. Fractions enriched for the POI were combined, supplemented with 1/20 (w/w) TEV protease and dialyzed against dialysis buffer overnight. The sample was then diluted to 100 mM NaCl and loaded onto a HiTrap Heparin HP 5 ml column (Cytiva), pre-equilibrated with dialysis buffer containing 100 mM NaCl. After washing with dialysis buffer containing 100 mM NaCl, the POI was eluted with a linear gradient to dialysis buffer containing 1.5 M NaCl. The fractions containing the POI were pooled and concentrated with a centrifugal concentrator (10 kDa molecular mass cutoff). The concentrated sample was further purified by SEC on a Superdex 75 10/60 GL column (Cytiva) in 20 mM TRIS-HCl, pH 7.5, 250 mM NaCl, 1 mM DTT. Fractions containing the POI were combined, concentrated, aliquoted, flash-frozen in liquid nitrogen and stored at −80 °C.

### Analytical size-exclusion chromatography
Analytical SEC-based interaction tests were conducted in 20 mM HEPES-NaOH, pH 7.5, 250 mM NaCl, 5% (v/v) glycerol, 1 mM DTT. In total, 100 pmol of ASCC3$^{HR}$ were mixed with other proteins in a two to tenfold molar excess in a final reaction volume of 80 μl. After incubation of the mixtures on ice for 30 min, the samples were loaded on a Superdex 200 3.2/300 analytical size-exclusion column (Cytiva). Overall, 50 μl fractions were collected and subjected to SDS-PAGE analysis. Protein bands were visualized by Coomassie staining except for gels containing TRIP4$^{1-80}$ or TRIP4$^{152-230}$, which were imaged by silver staining.

For testing competitive binding of TRIP4 and ALKBH3 to ASCC3$^{HR}$, 120 pmol of ASCC3$^{HR}$ (or of pre-formed ASCC3$^{HR}$-TRIP4 complex) were mixed with 360 pmol each of TRIP4 and ALKBH3 (or of ALKBH3) in a volume of 100 μl. After 30 min of incubation on ice, the samples were loaded on a Superdex 200 3.2/300 analytical size-exclusion column. In total, 50 μl fractions were collected and subjected to SDS-PAGE analysis. The proteins were visualized by Coomassie staining.

### DNA and RNA unwinding assays
DNA duplex unwinding activity was assessed in fluorescence-based stopped-flow experiments on a SX-20MV spectrometer (Applied Photophysics)[36,37]. The DNA substrate contained a 12-base pair duplex region and a 31-nucleotide 3′-ss overhangs, with an Alexa 488 fluorophore on the short strand and an Atto 540 Q quencher on the complementary strand ([Atto 540 Q]5′-**GGCCGCGAGCCG**GAAATTTAATTATAAACCAGACCGTCTCCTC-3′; 5′-**CGGCTCGCGGCC**-3′[Alexa 488]; duplex region in bold). Reactions were carried out in 40 mM HEPES-NaOH, pH 7.5, 80 mM NaCl, 0.5 mM MgCl$_2$ at 30 °C. 250 nM protein or protein complex were pre-incubated with 50 nM DNA

duplex for 5 min. Overall, 60 μl of the protein–DNA mixture were rapidly mixed with 60 μl of 4 mM ATP/MgCl$_2$, and the excited Alexa 488 fluorescence signal was recorded for 20 min using a 495 nm cutoff filter (KV 495, Schott). Selected experiments were repeated, by rapidly mixing 60 μl of the protein–DNA mixture with 60 μl of a solution containing 4 mM ATP/MgCl$_2$ and 2500 nM of the unlabeled short DNA strand as a trapping oligodeoxynucleotide (50-fold molar excess over labeled duplex). For each experiment, at least two individual traces were averaged, baseline-corrected by the fluorescence immediately after the addition of ATP and normalized to the baseline-corrected maximum fluorescence of the highest-amplitude trace of an experimental series. Data for ASCC3$^{HR,D611A}$-based unwinding had been reported previously[10] and are reproduced here to facilitate direct comparison. Data were plotted using Prism (version 9.0; GraphPad) and fitted to a double exponential equation (fraction unwound = $A_{fast}*(1 − \exp(−k_{fast} * t)) + A_{slow} * (1 − \exp(−k_{slow} * t))$; $A_{fast/slow}$, unwinding amplitudes of the fast/slow phases; $k_{fast/slow}$, unwinding rate constants of the fast/slow phases [s$^{-1}$]; $t$, time [s])[35]. Amplitude-weighted unwinding rate constants were calculated as $k_{uaw} = (A_{fast} * k_{fast}^2 + A_{slow} * k_{slow}^2) / (A_{fast} * k_{fast} + A_{slow} * k_{slow})$[35].

RNA duplex unwinding activities were assessed in the same way, using an RNA substrate with sequences analogous to the employed DNA substrate ([Atto 540 Q]5′-**GGCCGCGAGCCG**GAAAUUUAAUUAUAAACCAGACCGUCUCCUC-3′; 5′-**CGGCUCGCGGCC**-3′[Alexa 488]; duplex region in bold). For RNA unwinding assays, the excited Alexa 488 fluorescence signal was recorded for 90 min.

### ATPase assays
Thin layer chromatography (TLC)-based ATPase assays were performed using [α-$^{32}$P]ATP (Hartmann Analytic)[36,37]. To quantify DNA-stimulated ATPase activity, 0.5 μM protein or protein complex were combined with 1 mM of a 43-nucleotide ssDNA (5′-GGCCGCGAGCCG-GAAATTTAATTATAAACCAGACCGTCTCCTC-3′). 0.5 μM protein or protein complex or equivalent protein–DNA mixtures were incubated with 1 mM [α-$^{32}$P]ATP in 50 mM HEPES-NaOH, pH 7.5, 80 mM NaCl, 5 mM MgCl$_2$, 2 mM DTT at 30 °C for up to 60 min. 5 μl of sample were withdrawn at selected time points and reactions were quenched with 5 μl of 100 mM EDTA. 0.8 μl of the samples were spotted on a PEI-cellulose TLC plate and chromatographed with 1 M acetic acid, 0.5 M LiCl, 20 % (v/v) ethanol. The corresponding ADP and ATP spots were visualized using a Storm 860 phosphorimager (GMI, USA) and quantified using ImageQuant software (version 5.2; Cytiva). Data were plotted and analyzed using Prism (version 9.0; GraphPad), the ATPase activity was calculated as the number of hydrolyzed ATP molecules per protein molecule per minute, by fitting quantified data to the equation $V = (A_{fast} * V_{fast}^2 + A_{slow} * V_{slow}^2)/(A_{fast} * V_{fast} + A_{slow} * V_{slow})$; $A_{fast/slow}$, amplitudes of the fast/slow hydrolysis phases; $V_{fast/slow}$, rates of the fast/slow hydrolysis phases [min$^{-1}$]; $V$, ATP hydrolyzed as a function of time [min$^{-1}$].

### Microscale thermophoresis
Protein–DNA affinities were analyzed via microscale thermophoresis (MST) on a Monolith NT.115 instrument (NanoTemper) in standard capillaries. Protein solutions were centrifuged at 15,000×*g* for 5 min to remove aggregates and used to prepare 13- or 16-sample serial dilutions (1.22 nM to 5 μM for ASCC3$^{HR}$ and ASCC3$^{HR}$-TRIP4; 1.22 nM to 40 μM for TRIP4 and TRIP4$^{1-230}$; 0.47 nM to 1.94 μM for ASCC3$^{HR}$-TRIP4$^{1-230}$) in 20 mM HEPES-NaOH, pH 7.5, 300 mM NaCl, 5% (v/v) glycerol, 1 mM DTT. In all, 5 μl of a solution containing 100 nM [Cy5]5′-T$_{48}$ DNA oligomer were then added to 5 μl of each protein or protein complex solution and incubated at room temperature for 10 min. MST measurements were carried out at 40 % excitation power, 20% MST power and laser off/on times of 0/30 s. All experiments were repeated at least four times. Normalized fluorescence values ($F_{norm}$) were calculated as the ratios of the fluorescence values in the heated state (2.5 s

after IR laser heating) to the fluorescence values in the cold state (before laser heating). Dose/response curves were obtained by plotting $F_{norm}$ values after subtraction of baseline values ($\Delta F_{norm}$) against the logarithm of the protein or protein complex concentration. Interactions were quantified by data fitting to a Hill model ($F_{norm} = F_{norm,max} * X^h / (EC50^h + X^h)$; $F_{norm,max}$, maximum $F_{norm}$ value; $X$, concentration of the protein or protein complex; $h$, Hill slope; $EC50$, concentration needed to achieve a half-maximum binding at equilibrium) using Prism (version 9.0; GraphPad).

## Fluorescence microscopy

The sub-cellular localizations of the Flag/His-tagged versions of TRIP4 were determined by immunofluorescence[51]. 293T cells expressing Flag-tagged TRIP4 variants were grown on coverslips and fixed using 4% (v/v) paraformaldehyde for 20 min before permeabilization using 0.1% (v/v) Triton-X-100 in PBS for 20 min. Cells were blocked using PBS supplemented with 10% (v/v) fetal bovine serum (FBS) and 0.1% (v/v) Triton-X-100 for 1 h, then treated for 2 h with an FITC-conjugated α-Flag-M2 antibody (Sigma-Aldrich F4049; 1:200) diluted in PBS containing 10% FBS and 0.1% Triton-X-100. Cells were washed, and coverslips were mounted using mounting media containing DAPI. Cells were imaged using a Nikon Ti2 2-E inverted microscope.

## Immunoprecipitation and western blotting

293T cells expressing N- or C-terminally Flag/His-tagged versions of full-length or truncated TRIP4 or the Flag tag were lysed by sonication in IP buffer (50 mM Tris-HCl, pH 7.4, 150 mM NaCl, 0.5 mM EDTA, 0.1% (v/v) Triton-X-100, 10% (v/v) glycerol and cOmplete™ protease inhibitors. Lysates were cleared of debris by centrifugation at 20,000×$g$ for 10 min, then the cleared lysates were incubated with α-Flag-M2 magnetic beads (Sigma-Aldrich M8823) for 2 h. The matrix was washed five times with IP buffer and complexes were eluted using 3xFlag peptide (Sigma-Aldrich SAE0194). Proteins were precipitated using 20% (w/v) trichloroacetic acid (TCA) and separated by SDS-PAGE. Western blotting was performed using antibodies against the Flag tag (Sigma-Aldrich F3165; 1:7500), ASCC1 (Proteintech 12301-1-AP; 1:500), ASCC2 (Proteintech 11529-1-AP; 1:1000) and ASCC3 (Proteintech 17627-1-AP; 1:1000).

For immunoprecipitation of HA-tagged TRIP4 variants (TRIP4^wt, TRIP4^ΔZnF, TRIP4^L174A-L180A-I190A or TRIP4^C171A-C184A), the transfected 293 T cells were re-suspended in ice-cold, high salt co-IP buffer (50 mM Tris-HCl, pH 7.9, 300 mM KCl, 10% [v/v] glycerol, 1% [w/v] Triton-X-100, 1 mM DTT) supplemented with protease inhibitors. The cells were then lysed by sonication and allowed to rotate at 4 °C to complete lysis. Lysates were cleared by centrifugation and diluted to 150 mM KCl using co-IP buffer without KCl. α-HA beads (Santa Cruz Biotechnology, sc-7392 AC) were then added to the samples, and after incubation at 4 °C for 3.5 h, the beads were centrifuged and washed multiple times with 150 mM KCl co-IP buffer. Bound proteins were eluted with SDS-PAGE loading buffer and boiled before analysis via SDS-PAGE/western blot using antibodies against the HA-tag (Abcam EPR22819-101, 1:4000) and ASCC3[9].

## MMS sensitivity assays

The wt and TRIP4 KO PC-3 cells were plated on a 96-well plate with 3500 cells per well. Cells were exposed to media containing variable concentrations of MMS for 24 h at 37 °C. Then, cells were recovered with fresh culture medium for an additional 48 h at 37 °C. Cell viability was measured by using the MTS assay (Promega).

## Cryogenic electron microscopy

The ASCC3^HR-TRIP4 complex was prepared freshly in buffer 20 mM HEPES-NaOH, pH 7.5, 300 mM NaCl, 1 mM DTT, and concentrated to 4.15 mg/ml using a 50k ultra centrifugal filter (Merck). The sample was

supplemented with 0.01% (w/v) n-dodecyl β-maltoside promptly before vitrification. 3.8 µl of the sample were applied to glow-discharged holey carbon R1.2/1.3 copper grids (Quantifoil Microtools, Germany) and plunge-frozen in liquid ethane using a Vitrobot Mark IV (Thermo Fisher Scientific) equilibrated at 10 °C and 100% humidity.

Data acquisition was conducted on a FEI Titan Krios G3i TEM operated at 300 kV equipped with a Falcon 3EC detector. Movies were taken for 40.57 s accumulating a total electron flux of ~40 el/Å$^2$ in counting mode at a calibrated pixel size of 0.832 Å/px distributed over 33 fractions.

## CryoEM data analysis

All image analysis steps were done with cryoSPARC (version 3.2.2)[52]. Movie alignment was done with patch motion correction generating Fourier-cropped micrographs (pixel size 1.664 Å/px), CTF estimation was conducted by Patch CTF. Class averages of manually selected particle images were used to generate an initial template for reference-based particle picking from 6022 micrographs. In total, 2,818,857 particle images were extracted with a box size of 160 px and Fourier-cropped to 80 px for initial analysis. The reference-free 2D classification was used to select 1,590,881 particle images for further analysis. Ab initio reconstruction using a small subset of particles was conducted to generate an initial 3D reference for consecutive iterations of 3D heterogeneous refinement. Overall, 597,971 particle images were re-extracted with a box of 160 px and subjected to non-uniform refinement followed by CTF refinement. Another heterogeneous refinement round was applied to select 473,863 particle images for re-extraction at full spatial resolution after local motion correction (box size 320 px, 0.832 Å/px). A final heterogeneous refinement run was conducted to select 244,064 particle images for non-uniform refinement and generate the final reconstruction at a global resolution of 3.4 Å, locally extending down to 2.5 Å.

## Model building, refinement, and analysis

AlphaFold-predicted models[34] of ASCC3^HR and of regions of TRIP4 were manually placed in the cryoEM reconstruction and adjusted by rigid body fitting and segmental real-space refinement using Coot (version 0.9.8.1)[53]. The model was refined by iterative rounds of real-space refinement in PHENIX (version 1.20_4459)[54] and manual adjustment in Coot. Manual adjustments also took advantage of locally refined, focused cryoEM reconstructions. The structural model was evaluated with Molprobity (version 4.5.1)[55]. Interface areas were analyzed via the PISA server (version 1.52)[56]. Structure figures were prepared using ChimeraX (version 1.4)[57] and PyMOL (version 1.8; Schrödinger, LLC).

## DNA−protein cross-linking/mass spectrometry

UV-cross-linking was employed to generated zero length cross-links between protein and bound ssDNA oligos ($T_{12}$, $T_{24}$, $T_{36}$, $T_{48}$). DNA oligos were 5′-end labeled using [γ-$^{32}$P]ATP and T4 polynucleotide kinase using a standard protocol. In all, 10 µl reaction mixtures containing 100 nM ("1" in Fig. 4b) or 200 nM ("2" in Fig. 4b) protein or protein complex and 4.3 nM radio-labeled DNA probe were incubated in a 72-well microbatch plate (Greiner) in 50 mM HEPES-NaOH, pH 7.5, 80 mM NaCl, 5 mM MgCl₂, 2 mM DTT on ice for 5 min, then the samples were exposed to 254 nm UV irradiation for 10 min (Ultraviolet cross-linker, Amersham Life Science). Cross-linked samples were separated by SDS-PAGE and visualized by autoradiography using a Storm 860 phosphorimager.

For identifying cross-linked peptides and residues, 6.7 nM unlabeled $T_{48}$ ssDNA were cross-linked to 200 nM ASCC3^HR or ASCC3^HR-TRIP4 in 48 × 10 µl reactions as above and ethanol precipitated. Subsequent analyses were conducted in duplicates. The pellets were

dissolved in 50 µl 4 M urea and diluted to 1 M urea with 50 mM Tris-HCl, pH 7.5. To digest the DNA, 1 µl Universal nuclease (Pierce) and 1 µl Nuclease P1 (New England Biolabs) were added to the samples, followed by incubation at 37 °C for 3 h. Protein digestion was performed with 1 µg of trypsin (Promega) overnight at 37 °C. The samples were acidified with formic acid (FA; final concentration 0.1% [v/v]), and acetonitrile (ACN) was added to 5% (v/v) final concentration. Non-cross-linked nucleotides were depleted by C18 reversed-phase chromatography with Harvard Apparatus MicroSpin columns. The sample was eluted by stepwise application of 50% (v/v) and 80% (v/v) ACN. Cross-linked peptides were enriched over linear peptides by TiO2 self-packed tip columns with 5% (v/v) glycerol as a competitor[58]. The samples were dried under vacuum and re-suspended in 10 to 15 µl of 2% (v/v) ACN, 0.05% (v/v) trifluoroacetic acid. Seven or 8 µl (first or second analysis) were used for LC-MS analysis.

Chromatographic separation was achieved with Dionex Ultimate 3000 UHPLC (Thermo Fischer Scientific) coupled with a C18 column packed in-house (ReproSil-Pur 120 C18-AQ, 1.9/3 µm particle size, 75 µm inner diameter, 30 cm length, Dr. Maisch GmbH). The flow rate was set to 300 nl/min, and a 44 min linear gradient was formed with mobile phase A (0.1% [v/v] FA) and B (80% [v/v] ACN, 0.08% [v/v] FA) from 8% or 10% (first or second analysis) to 45% mobile phase B. Data acquisition of eluting peptides was performed with Orbitrap Exploris 480 (Thermo Fischer Scientific). The resolution for survey scans was set to 120,000, the maximum injection time to 60 ms, the automatic gain control target to 100% or 250% (first or second analysis), and the dynamic exclusion to 9 s. Analytes selected for fragmentation were isolated with a 1.6 $m/z$ window and fragmented with a normalized collision energy of 28. MS/MS spectra were acquired with a resolution of 30,000, a maximum injection time of 120 ms, and an automatic gain control target of 100%.

Cross-link data analysis of the resulting raw files was performed with the OpenNuXL node of OpenMS (version 3.0.0)[59]. Default general settings were used and the preset DNA-UV Extended was selected. The sequences of the proteins in the sample were provided as a database. The maximum length of DNA adducts was set to 3, and poly-T was used as sequence. The resulting.idxml files were used for annotation, and spectra were manually validated.

### Reporting summary
Further information on research design is available in the Nature Portfolio Reporting Summary linked to this article.

## Data availability
The cryoEM reconstruction of the ASCC3^HR-TRIP4 complex has been deposited in the Electron Microscopy Data Bank (https://www.ebi.ac.uk/pdbe/emdb) under accession code EMD-15521. Structure coordinates have been deposited in the RCSB Protein Data Bank (https://www.rcsb.org) with accession code 8ALZ[60]. The DNA–protein CLMS data have been deposited in the ProteomeXchange Consortium (http://www.proteomexchange.org) via the PRIDE[61] partner repository (https://www.ebi.ac.uk/pride/) under dataset identifier PXD036106. All other data are contained in the manuscript or the Supplementary Information. Structure coordinates used in this study are available from the RCSB Protein Data Bank (https://www.rcsb.org) under the accession codes 2P6R and 4F91. Source data are provided with this paper.

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

## Acknowledgements

We thank Agnieszka Pietrzyk-Brzezinska (Lodz University of Technology, Poland) for help in the cloning of the ALKBH3 expression construct, Philipp Hackert (University Medical Center Göttingen, Germany) for technical assistance, Ralf Pflanz and Monika Raabe (Max-Planck-Institut für Multidisziplinäre Naturwissenschaften, Germany) for help in mass spectrometric analysis and Daniel Lauster (Freie Universität Berlin, Germany) for advice in MST analyses. We acknowledge the assistance of the core facility BioSupraMol supported by the Deutsche Forschungsgemeinschaft in electron microscopic analyses. This work was supported by grants from the Deutsche Forschungsgemeinschaft (INST 130/1064–1 FUGG to Freie Universität Berlin; SFB1565 [project number 469281184], to K.E.B., M.T.B., H.U., and M.C.W.; BO 3442/1-2 [project number 192916677] to M.T.B.; SFB860 [project number 105286809] to K.E.B. and H.U.), the American Cancer Society (RSG –18–156-01-DMC to N.M.), the National Institutes of Health of the

U.S. (R01 CA193318 and P01 CA092584 to N.M.) and the Berlin University Alliance (501_BIS-CryoFac to M.C.W.).

## Author contributions

J.J. cloned genes, purified proteins, assembled complexes for cryoEM and CLMS, conducted ATPase assays, and generated stable cell lines. J.J., A.A., and N.H. performed in vitro interaction assays and unwinding experiments. T.H. acquired, processed, and refined cryoEM data. J.J. and B.L. built and refined atomic models. K.E.B. generated stable cell lines for the inducible expression of Flag-tagged TRIP4 variants and conducted cellular localization and pull-down analyses. L.P. generated stable cell lines for the expression of HA-tagged TRIP4 variants and conducted pull-down analyses. N.T. created TRIP4 KO cells and analyzed MMS sensitivity. A.C. and J.B. conducted DNA–protein CLMS analyses. All authors contributed to the analysis of the data and the interpretation of the results. J.J. and M.C.W. wrote the manuscript with contributions from the other authors. N.M., M.T.B., H.U., and M.C.W. supervised work in their respective groups and coordinated the collaboration.

## Funding

## Competing interests

The authors declare no competing interests.
