## [Peer Review File · Nature Communications]

Extended DNA threading through a dual-engine motor module of the activating signal co-integrator 1 complexREVIEWER COMMENTS

Reviewer #1 (Remarks to the Author):

In the present manuscript, the authors focus on the biochemical and structural characterization of the helicase ASSC3. The structural and biochemical data are of high quality and very interesting. Still, I think it would be essential to perform DNA binding assays to gain insights into the mechanism of helicase stimulation.

I have to say that I found reading the manuscript challenging at several points. The complex subunits have cryptic names, and ASC1 and ASCC1 can be easily confused, but this is not the author's fault. The introduction reads like a compendium list that lists all the functions the complex has been associated with. For this paper, it is not essential that A4220 of 28S rRNA has been modified in a process that requires ASSC. In the current form, the introduction does not prepare the reader for the findings in the results section - regulation of helicases and the observation that both helicase domains are active and seem to cooperate. Also, the essential and interesting information about the yeast ASSC3 ortholog has only been mentioned in the discussion. In my opinion, a mechanistic and structural understanding of the yeast ortholog should be mentioned in the introduction. It would be great if the authors could have a look at the introduction.

The results section starts with the statement that ASSC3 binds both ASCC1 and ASCC2, but then ASC1 is investigated further. This is confusing – maybe the authors wanted to write that ASSC3 binds ASC1 and ASCC2?

From here on, the manuscript focuses on the ASSC3/ASC1 complex, but ASCC2 is never mentioned again. As a reader, I would like to know why the authors did not study the trimeric complex. It would be great if the author could explain their focus on the ASSC3/ASC1 subcomplex.

The structure in Figure 2a is not intuitive to understand as it is not clear what part of the map belongs to ASSC3 and which to ASC1. I am an experienced structural biologist, but I would suggest trying different ways of presentation to see what works best so that the reader can immediately grasp what the helicase is and what ASC1 is. For example, maybe color the NC in light blue, the CC in dark blue, and the ASC1 can stay in variations of yellow/brown. I would already, at this point, highlight the helicase entry sites.

I appreciate the careful analysis and testing of both N and C tagging, but I would simplify figure 3 and make it smaller. I have three suggestions:

- i) Focus on the IP data and then move the IF data to the supplement with the comment that truncations of ASC1 don't affect the subcellular localization.
- ii) Focus on either the N or the C-terminal tagging approach. Show for one of them both IF and IP and move the other to the supplement with the comment that the localization of the Flag Tag does not affect the experiment.
- iii) Keep the IP data for N tagging as panel A and the ASC1 mutants as panel B – this has all the information needed for the reader!

When the authors switched to the functional characterization of the helicase function, the authors made several statements that would benefit from additional explanations.

"As this assay tested multiple rounds of unwinding, the observed time traces were fit to a double exponential equation, and amplitude-weighted unwinding rate constants (k_{uaw}) were calculated... ."

In the assay, the helicase is 5-fold more concentrated than the substrate – approaching single turnover conditions. What causes the multiple turnover conditions here? Could the helicase DNA ratio be changed? The signal-to-noise ratio is good, so decreasing DNA and increasing the helicase should be possible.

Or is the multiple turnover scenario caused by the reannealing of the ssDNA, which regenerates the

dsDNA substrate? Could the authors use a trapping strand that prevents this?
It would also be great if the authors could speculate why there are a slow and a fast phase.
Interestingly, both rate constants are increased two-fold by ASC1, even with the inactive CC.

The most interesting question will be how the two active helicase domains act in concert. The CC domain seems less important than the NC domain as the CC mutant can still unwind dsDNA, even though the helicase activity is reduced compared to WT. Unfortunately, it is not known if the decrease of helicase activity results in a phenotype in cells. The authors mention that in Slh1p, the NC was mutated to trap the RQT-ribosome complex. What happens if the mutation is introduced in CC of Slh1p? Could that be assessed in yeast if this is not accessible in human cells?

The authors show that ASC1 activates the helicase activity without stimulating the ATPase. The authors suggest that the presence of the ASCH domain might facilitate DNA exit. Interestingly, ASC1 can activate the CC mutant - is it possible that the DNA is threaded through the CC even though this is inactive? Or how do the authors envisage this?

It would be great if the authors would assess DNA binding affinities of ASCC3 in the absence or presence of ASC1 and with ASC1 lacking the ASCH domain. This would provide essential data underpinning the mechanisms proposed by the authors.

For me, the last section of the manuscript that analyzes the interaction with ALKBH3 broke the flow because we zoom out again from the mechanism of unwinding to the analysis of a different interaction partner, ALKBH3. Interestingly, ALKBH3 competes with ASC1 for ASCC3 binding. But I would consider including this finding in the initial mapping experiments around figure 1 and conclude with the statement that the helicase core is an interaction hub that can bind both ASC1 and ALKBH3 – but ALKBH3 seems to bind weaker than ASC1. Then one could switch to the structure of the ASCC3/ASC1 complex.

As both ALKBH3 and ASC1 compete for binding, the obvious question is: Do they both stimulate the helicase activity? Could this be tested? Because it would be an important finding if different binding partners affect the helicase function similarly or differently.

Minor:

The authors mapped the interacting regions by SEC. Is there a reason why the SEC profiles are not shown?

It would be great if in Figure 4B the WT could be plotted again so that one can visually inspect the difference in the point mutants.

In Figure 5a, I have problems seeing the path for the DNA. The figure might suffer from the PDF conversion, but also, the electrostatic surface potential makes it challenging to see. The ssDNA region from the Hel308 structure only features 9 nucleotides – in figure 5a 24 nts, how was this modeled?

Figure 5b: DNA T24 is a defined band but for T36 and T48 a band resembling T24 is also present. Are the DNAs degraded or?

In the results section, the MMS assay is not explained – it might be worth adding what MMS does and how this connects to ALKBH3.

Reviewer #2 (Remarks to the Author):

The manuscript "Extended DNA threading through a dual-engine motor module in the activating signal co-integrator complex" by Jia, et al, describes the structure and activity of the ASCC3 subunit of the human activating signal co-integrator complex (ASCC). ASCC plays roles in transcriptional regulation, DNA repair and ribosome quality control. ASCC3 (the primary focus of this study) is a large Ski2-like DNA helicase/translocase that contains 2 helicase cassettes (an N-terminal cassette and a C-terminal cassette). In other characterized Ski2-like tandem-cassette helicases, only the N-terminal cassette exhibits helicase activity. However, in this study, the authors demonstrate that both cassettes of ASCC3 are functional. A cryo-EM structure shows how the cassettes pack together. Modeling of a DNA substrate onto the structure, combined with cross-linking/mass spectroscopy analysis, suggests that the DNA moves directly from the N-terminal to the C-terminal cassette.

Additionally, ASCC1, an activator of ASCC3 helicase activity, is shown to interact directly with the C-terminal cassette of ASCC3 through the ZnF domain using a combination of deletion mutants and analytical size-exclusion chromatography. The binding interaction is captured in the cryo-EM structure.

This is a thorough and well-written manuscript that makes an important contribution to the field. The observation that both cassettes are functional is particularly interesting and will be of interest to the broader helicase community. I have no concerns with the substance of the manuscript. However, the authors could consider the following comments:

It is noted in the Discussion that ASCC3 exhibits unwinding activity on RNA substrates, although at a significantly reduced level compared to DNA (data not shown). Can the authors provide data to demonstrate how activity compares between RNA and DNA?

Fig 1. While I have no concerns with the conclusions presented by the authors, the layout of the cropped gels in Fig 1 are somewhat disorienting – specifically when the width of the gels differs in a single panel (e.g. panel b). Additionally, the cropping of the final two gels in panel e appears to be a bit aggressive. The full gels included as source data provide essential and reassuring context. I don't have specific suggestions for how to improve the figure, but I encourage the authors to consider adjustments to improve readability.

Fig 5. Panel a and d. It isn't clear what the boxed rotations refer to. My guess is that it refers to rotations compared to Fig 2, but I don't see it specified in the figure or the figure legend. Also, "orthogonal" (pg 40 ln 917) generally refers to right angles. I think the figure is showing a 180 deg rotation instead.

Minor comments:

Pg 12 ln 317. Remove comma after "mechanism"

Pg 13 ln 359. Remove comma after "possible"

Pg 16 ln 434. Should be "pre-equilibrated"

Reviewer #3 (Remarks to the Author):

Jia J. et al report the structural and biochemical characterization of the activating signal co-integrator complex (ASCC) subunit 3 in complex with ASC1. The authors show that both helicase domains of ASCC3 are active in contrast to the homologue Ski2-like helicase SNRNP200. The data further indicate that ASC1 and ALKBH3 binding to ASCC3 is mutually exclusive.

Although the structure of ASCC3 in complex with ASC2 has been solved before by the authors, the interaction with the regulatory ASC1 subunit was unknown before and gives important insight into the regulation of the ASCC3 helicase.

The data are detailed and solid and proof all conclusions drawn.

However, I have some suggestion to improve the manuscript.

For readers from outside the field it is difficult to understand which ASCC proteins form a stable complex, and which are auxiliary proteins. For example, are ASCC1 and ASCC2 constitutively bound to ASCC3? Where would ASCC2 interact in the ASCC3-ASC1 complex or is the interaction also mutual exclusive?

Maybe the comparison of the ASCC3-ASCC2 and ASCC3-ASC1 complex would help to understand the regulation?

1)Page 2 Line 29: What does it mean that ASCC3 is an unconventional helicase?

2) Page 3, line 55-60: The sentence is too long and barely understandable for people outside the field.

3) It would be better to put some of the western blots in Figure 1 (c, e, f) in the supplementary data and show the structures from figure 2a and b in Figure 1 instead. This is necessary to immediately understand the color code in Fig. 2 a,b) Figure 2 c has a completely different color code and Figure 2d fits to a and b again. This is a bit confusing. Further the numbering and the interacting residues in Figure 2d (first image) are barely visible.

4) The information content of the DNA binding model in Figure 5a is rather low and it somehow looks like the DNA is bound by negatively charged residues. Why is the DNA chain broken in Figure 5d? Could you please optimize this model or illustrate it better?

5)Just out of curiosity: The protein -DNA crosslink looks very efficient - what was the problem with the cryo EM structure of the DNA bound complex. Did you try different grid types?

6) Fig. 3c (Input) and Fig. 5b some western blots look like modified in contrast and brightness too much. Could you please provide an unmodified image?

Response to Reviewer Comments

Reviewer comments are repeated in bold italics, responses are in regular font, changed text passages are highlighted in yellow.

Please note:

- As pointed out by Reviewer 1, the nomenclature for subunits of protein complexes relevant to the present manuscript is somewhat confusing. To alleviate this problem and to avoid confusion of ASC1 and ASCC1, we decided to refer to the ASC1 protein by its alternative name, TRIP4, in the revised manuscript.
- Line numbers quoted below refer to the combined manuscript and supplement file with marked up changes submitted as a “Related Manuscript File”.

Reviewer #1:

In the present manuscript, the authors focus on the biochemical and structural characterization of the helicase ASSC3. The structural and biochemical data are of high quality and very interesting. Still, I think it would be essential to perform DNA binding assays to gain insights into the mechanism of helicase stimulation.

We thank the reviewer for the very positive overall evaluation and for considering our study to be of high quality and very interesting. As suggested, we have conducted additional DNA binding analyses as further detailed below.

I have to say that I found reading the manuscript challenging at several points. The complex subunits have cryptic names, and ASC1 and ASCC1 can be easily confused, but this is not the author’s fault. The introduction reads like a compendium list that lists all the functions the complex has been associated with. For this paper, it is not essential that A4220 of 28S rRNA has been modified in a process that requires ASCC. In the current form, the introduction does not prepare the reader for the findings in the results section - regulation of helicases and the observation that both helicase domains are active and seem to cooperate. Also, the essential and interesting information about the yeast ASSC3 ortholog has only been mentioned in the discussion. In my opinion, a mechanistic and structural understanding of the yeast ortholog should be mentioned in the introduction. It would be great if the authors could have a look at the introduction.

We agree with the reviewer’s assessment. To somewhat alleviate the problem of confusing protein nomenclature, we now consistently refer to “ASC1” by its alternative name “TRIP4”, to more clearly distinguish it from the “ASCC1” protein. Also as suggested, we thoroughly revised the introduction. To properly set the stage, we now start with a brief section on nucleic acid-dependent NTPases/helicases (line 45):

Nucleic acid-dependent nucleoside-triphosphatases (NTPases) are pervasively involved in processes related to DNA replication, recombination, genome maintenance, gene expression and co-/post-transcriptional gene regulation.¹ These enzymes exhibit nucleic acid binding, translocating and/or unwinding activities, and are often referred to as DNA or RNA helicases, depending on the nucleic acid specificity.^{2,3} *In vivo* they utilize to these molecular activities to often act as versatile remodelers of nucleic acid-protein complexes.^{4,5} The intrinsic molecular mechanisms of nucleic acid-dependent NTPases are diverse, relying on core RecA-like NTPase domains that frequently are functionally expanded by peripheral regions and auxiliary domains, and can be further modulated by interacting regulators.^{5,6} Several nucleic acid-dependent NTPases are involved in more than one cellular process, affording the potential for functional coordination and cross-regulation.^{7,8}

Activating signal co-integrator 1 complex (ASCC) subunit 3 (ASCC3) has been characterized as a 3'-to-5' directional DNA translocase/helicase^{9,10}...

We then introduce ASCC3 as a particularly versatile, multi-functional helicase with a domain composition only shared by the spliceosomal RNA helicase, SNRNP200. Subsequently, we illustrate the diverse molecular functions that ASCC3 is involved in, presumably as a subunit of distinct complexes with overlapping but non-identical subunit composition. In the brief section on ASCC3-linked translation quality control, we also follow the reviewer's advice to briefly introduce its yeast ortholog, Slh1p, and orthologs of other subunits of the respective complex (as translation/ribosome quality control is the main ASCC3/Slh1p function that has been characterized in yeast). We then point out remaining key questions regarding the molecular mechanism and mode of regulation of ASCC3 that are likely relevant in all its functional contexts, finishing up with a brief summary of our key findings. We left out information not directly pertaining to the present topic, such as ASCC's potential role in 28S rRNA modification or P-body formation.

The results section starts with the statement that ASSC3 binds both ASCC1 and ASCC2, but then ASC1 is investigated further. This is confusing – maybe the authors wanted to write that ASSC3 binds ASC1 and ASCC2?

Our intention was to start the Results section with a statement that calls attention to the fact that ASCC3 not only fulfills a function as a nucleic acid helicase but also as a scaffold for the complexes it is involved in. To stick to this strategy but avoid confusion, we now start with a more general statement that avoids mention of proteins that we do not subsequently focus on (line 121):

Previous interaction mapping suggested that, apart from providing ATPase/helicase activities, ASCC3 may represent a major scaffold for assembling complexes for diverse cellular functions.^{9,10,13,31} We therefore tested whether TRIP4, which is implicated with ASCC3 in transcription regulation and ribosome quality control, also directly binds ASCC3 *in vitro*.

From here on, the manuscript focuses on the ASCC3/ASC1 complex, but ASCC2 is never mentioned again. As a reader, I would like to know why the authors did not study the trimeric complex. It would be great if the author could explain their focus on the ASCC3/ASC1 subcomplex.

We had previously shown that ASCC2 interacts with a small helical domain of the ASCC3 N-terminal region but not with the ASCC3 helicase region, elucidated the structural basis of the ASCC2-ASCC3 interaction and studied its relevance for ASCC3 auto-regulation (Jia *et al.*, *Nat Commun*, 2020). While we managed to obtain a ternary ASCC2-ASCC3-TRIP4 complex from recombinant components, we did not obtain a cryoEM reconstruction in which we could visualize all three components. Possibly the ASCC2-bound N-terminal region of ASCC3 remains flexible and multiply oriented relative to the TRIP4-bound ASCC3 helicase region in the isolated ternary complex. We also did not detect a direct, stable ASCC2-TRIP4 interaction *in vitro*. We therefore focused on the ASCC3^{HR}-TRIP4 interaction and its functional consequences for ASCC3 helicase activity in the present manuscript. We believe that with the new opening of the Results section that no longer refers to ASCC2 (see above) an explicit explanation of the above is perhaps not required in the revised manuscript.

The structure in Figure 2a is not intuitive to understand as it is not clear what part of the map belongs to ASSC3 and which to ASC1. I am an experienced structural biologist, but I would suggest trying different ways of presentation to see what works best so that the reader can immediately grasp what the helicase is and what ASC1 is. For example,

maybe color the NC in light blue, the CC in dark blue, and the ASC1 can stay in variations of yellow/brown. I would already, at this point, highlight the helicase entry sites.

We thank the reviewer for these suggestions to more clearly represent the structural organization of the complex and have adjusted former **Figure 2a,b,d** (new **Figure 1c-e**) accordingly. We have chosen to display the ASCC3^{HR} regions in different shades of gray and TRIP4 uniformly in red (as ZnF, lasso and ASCH regions are well separated and discernible in the views). Here the modified Figure for the reviewer's convenience:

I appreciate the careful analysis and testing of both N and C tagging, but I would simplify figure 3 and make it smaller. I have three suggestions:

- i) Focus on the IP data and then move the IF data to the supplement with the comment that truncations of ASC1 don't affect the subcellular localization.
- ii) Focus on either the N or the C-terminal tagging approach. Show for one of them both IF and IP and move the other to the supplement with the comment that the localization of the Flag Tag does not affect the experiment.
- iii) Keep the IP data for N tagging as panel A and the ASC1 mutants as panel B – this has all the information needed for the reader!

We followed suggestion iii) of the reviewer to minimize the former Figure 3 (now Figure 2 and Supplementary Figure 5). New Figure 2:

New Supplementary Figure 5:

When the authors switched to the functional characterization of the helicase function, the authors made several statements that would benefit from additional explanations. "As this assay tested multiple rounds of unwinding, the observed time traces were fit to a double exponential equation, and amplitude-weighted unwinding rate constants (kuaw) were calculated... ." In the assay, the helicase is 5-fold more concentrated than the substrate – approaching single turnover conditions. What causes the multiple turnover conditions here? Could the helicase DNA ratio be changed? The signal-to-noise ratio is good, so decreasing DNA and increasing the helicase should be possible.

Or is the multiple turnover scenario caused by the reannealing of the ssDNA, which regenerates the dsDNA substrate? Could the authors use a trapping strand that prevents this?

Time traces recorded under the stated conditions could only be adequately fitted by a double-exponential equation. As suggested by the reviewer, we had suspected that DNA strands can re-anneal after a first round of unwinding and can then be re-engaged and unwound again by ASCC3^{HR}. To test this notion, we now did control experiments, in which we included a trap oligo (identical to the short strand) in 50-fold excess. Corresponding time traces again could only be fitted with a double exponential equation, with very similar amplitudes and rate constants as in the absence of the trap. We therefore conclude that the biphasic unwinding behavior we observe is not the consequence of multiple rounds of unwinding due to DNA re-annealing, but rather an inherent characteristic of the ASCC3^{HR}-mediated unwinding process under our chosen conditions. In the revised manuscript, we therefore now omitted any reference to “multi-round conditions”. We describe the control experiments in the revised text (line 219):

To test if the biphasic unwinding behavior was due to multiple rounds of unwinding, we repeated the experiments for ASCC3^{HR} and the ASCC3^{HR}-TRIP4 complex in the presence of a 50-fold molar excess of a trapping DNA (unlabeled short strand of the duplex). Again, biphasic time traces yielding very similar rate constants as in the absence of a trap were observed (Supplementary Fig. 6a; Supplementary Table 2), suggesting that our initial assays did not monitor DNA re-annealing and re-binding of the helicase machineries. Instead, slow and fast phases may represent, e.g., an initial productive accommodation of DNA after ATP addition followed by the unwinding process *per se*, and both phases are stimulated by TRIP4.

The additional results are shown in the new Supplementary Figure 6a and in the expanded Supplementary Table 2:

Supplementary Table 2:

	ASCC3 ^{HR}	ASCC3 ^{HR} -TRIP4	ASCC3 ^{HR} -TRIP4 ¹⁻²³⁰	ASCC3 ^{HR} -TRIP4 ⁴⁰³⁻⁵⁸¹	ASCC3 ^{HR} ,D1453A	ASCC3 ^{HR} ,D1453A-TRIP4	ASCC3 ^{HR} -ALKBH3
Substrate	DNA minus trap						
$A_{fast}^{(a)}$	0.309	0.497	0.279	0.212	0.393	0.402	
$A_{slow}^{(a)}$	0.691	0.503	0.721	0.788	0.607	0.598	
$k_{fast} (s^{-1})^{(b)}$	0.031	0.061	0.038	0.045	0.013	0.028	

$k_{\text{slow}} (\text{s}^{-1})$ ^(b)	0.005	0.010	0.005	0.004	0.002	0.004	
$k_{\text{uaw}}^a (\text{s}^{-1})$ ^(c)	0.024	0.054	0.030	0.035	0.011	0.024	
R^2 ^(d)	0.999	0.998	0.998	0.998	0.995	0.996	
Substrate	DNA plus trap						
A_{fast} ^(a)	0.294	0.495					0.295
A_{slow} ^(a)	0.706	0.505					0.705
$k_{\text{fast}} (\text{s}^{-1})$ ^(b)	0.027	0.067					0.024
$k_{\text{slow}} (\text{s}^{-1})$ ^(b)	0.004	0.006					0.004
$k_{\text{uaw}}^a (\text{s}^{-1})$ ^(c)	0.021	0.062					0.018
R^2 ^(d)	0.999	0.998					0.998
Substrate	RNA minus trap						
A_{fast} ^(a)	0.010	0.103					
A_{slow} ^(a)	0.990	0.897					
$k_{\text{fast}} (\text{s}^{-1})$ ^(b)	0.097	0.052					
$k_{\text{slow}} (\text{s}^{-1})$ ^(b)	0.0003	0.001					
$k_{\text{uaw}}^a (\text{s}^{-1})$ ^(c)	0.073	0.046					
R^2 ^(d)	0.999	0.999					

It would also be great if the authors could speculate why there are a slow and a fast phase. Interestingly, both rate constants are increased two-fold by ASC1, even with the inactive CC.

We followed the reviewer's suggestion. Based on our additional control experiments and reasoning outlined above, we now included a brief corresponding statement (line 224):

Instead, slow and fast phases may represent, e.g., an initial productive accommodation of DNA after ATP addition followed by the unwinding process *per se*, and both phases are stimulated by TRIP4.

The most interesting question will be how the two active helicase domains act in concert. The CC domain seems less important than the NC domain as the CC mutant can still unwind dsDNA, even though the helicase activity is reduced compared to WT. Unfortunately, it is not known if the decrease of helicase activity results in a phenotype in cells. The authors mention that in Slh1p, the NC was mutated to trap the RQT-ribosome complex. What happens if the mutation is introduced in CC of Slh1p? Could that be assessed in yeast if this is not accessible in human cells?

We agree with the reviewer that these are interesting questions. While we believe that the proper assessment of functional consequences of ASCC3 variants containing separately inactivated NC or CC in cells, or the switch to similar analyses using the yeast system, is beyond the scope of the present manuscript, previous studies by other groups provide evidence for the functional importance of both ASCC3 cassettes as NTPases/helicases *in vivo*. Similar findings have been made in the yeast system with Slh1p. We had mentioned these findings in the previous version of our manuscript, but have now slightly expanded the corresponding section to clarify the situation (line 357):

In line with previous observations^{9,10}, our systematic ATPase and DNA unwinding assays strongly suggest that, in contrast to SNRNP200, both ASCC3 cassettes are active ATPases and helicases. Supporting this notion, previous studies suggested a role for ATPase activities in one or both cassettes of ASCC3 or its yeast ortholog, Slh1p, during ribosome quality control. E.g., reduced ribosome association was found for an ASCC3 variant with an ATPase-deficient NC²³, and selective abrogation of the ATPase activity in the ASCC3 NC or CC led to reduced disruption of stalled ribosomes²⁴. Likewise, the ATPase activity of the Slh1p NC was shown to be required for ribosome dissociation^{26,27,41} or completion of subunit splitting²⁵.

The authors show that ASC1 activates the helicase activity without stimulating the ATPase. The authors suggest that the presence of the ASCH domain might facilitate DNA exit. Interestingly, ASC1 can activate the CC mutant - is it possible that the DNA is threaded through the CC even though this is inactive? Or how do the authors envisage this?

Presently, we do not know if DNA is threaded through both helicase cassettes when the CC is inactivated or if the DNA takes advantage of an alternative exit site. The former possibility presently seems more likely considering that TRIP4 still has a stimulatory effect on CC-inactivated ASCC3^{HR}. Regarding further elucidation of the possible mechanisms of helicase activation by ASC1 (now referred to as TRIP4), we conducted additional experiments as suggested by the reviewer, please refer to the next point.

It would be great if the authors would assess DNA binding affinities of ASCC3 in the absence or presence of ASC1 and with ASC1 lacking the ASCH domain. This would provide essential data underpinning the mechanisms proposed by the authors.

We agree with the reviewer and have conducted additional DNA binding studies with ASCC3^{HR} and TRIP4 variants. We tested various types of assays and found multi-scale thermophoresis with fluorescence-labeled DNA to yield the clearest results in the present case. We find that both ASCC3^{HR} and TRIP4 can bind DNA individually and that the ASCC3^{HR}-TRIP4 complex exhibits increased DNA affinity compared to either protein alone. In contrast, the DNA affinity of a ASCC3^{HR}-TRIP4^{ΔASCH} complex (lacking the TRIP4 ASCH domain) was not enhanced compared to ASCC3^{HR} alone. We have included the additional DNA binding experiments (as new Figure 3e) and corresponding descriptions in the revised text (line 264):

TRIP4 enhances DNA engagement by ASCC3

The TRIP4 ASCH domain constitutes a putative nucleic acid-binding domain^{38,39} that might contribute to DNA binding by an ASCC3-TRIP4 complex. To test this notion, we conducted comparative DNA binding assays using a Cy5-labeled T₄₈ DNA oligomer in multi-scale thermophoresis (MST) assays. Both ASCC3^{HR} and TRIP4 individually bound T₄₈ DNA with an EC₅₀ of 551 ± 194 nM and 903 ± 24 nM, respectively (Fig. 3e). The T₄₈ DNA affinity of the ASCC3^{HR}-TRIP4 complex (EC₅₀ = 53 ± 3 nM) was ~ 10-fold stronger compared to isolated ASCC3^{HR}. In contrast, TRIP4¹⁻²³⁰ (containing the ZnF domain but lacking the lasso-like peptide and ASCH domain) exhibited very low DNA affinity (EC₅₀ n.d.) and did not significantly influence the DNA affinity of ASCC3^{HR} (EC₅₀ = 468 ± 63 nM). Therefore, additional nucleic acid contacts established *via* the TRIP4 ASCH domain could modulate the ASCC3^{HR}-mediated DNA unwinding mechanism and/or might influence initial substrate engagement by ASCC3^{HR}.

New Figure 3e:

e DNA binding

Regarding implications for the mechanisms of TRIP4-dependent helicase activation, we amended the Discussion section (line 389):

Our functional analyses suggest that TRIP4 could modulate the ASCC3 helicase by multiple, not mutually exclusive mechanisms, that seem to depend on nucleic acid binding by the ASCH domain^{38,39}. First, the DNA affinity of ASCC3^{HR} is enhanced in the presence of TRIP4 due to the ASCH domain. TRIP4 could, thus, support ASCC3 loading onto substrate DNA. Second, our DNA-protein CLMS data support the notion that the ASCH domain or neighboring regions may facilitate DNA exit from the ASCC3 CC, whereby TRIP4 might influence the ASCC3-mediated unwinding process, as observed.

For me, the last section of the manuscript that analyzes the interaction with ALKBH3 broke the flow because we zoom out again from the mechanism of unwinding to the analysis of a different interaction partner, ALKBH3. Interestingly, ALKBH3 competes with ASC1 for ASCC3 binding. But I would consider including this finding in the initial mapping experiments around figure 1 and conclude with the statement that the helicase core is an interaction hub that can bind both ASC1 and ALKBH3 – but ALKBH3 seems to bind weaker than ASC1. Then one could switch to the structure of the ASCC3/ASC1 complex.

We thank the reviewer for this suggestion and understand the reasoning behind this suggestion. However, as suggested by this reviewer (see next point), we have now also tested whether ALKBH3 modulates the helicase activity of ASCC3 and have included these results in the revised manuscript. Thus, the section on ALKBH3 refers to multiple aspects that we could not easily group at the beginning of the Results section. Rather than distributing them throughout the Results section, we decided in the end to keep them together as a final paragraph. We hope that the reviewer agrees.

As both ALKBH3 and ASC1 compete for binding, the obvious question is: Do they both stimulate the helicase activity? Could this be tested? Because it would be an important finding if different binding partners affect the helicase function similarly or differently.

Again, we agree and have conducted corresponding experiments. We find that unlike TRIP4, ALKBH3 has no influence on ASCC3^{HR} helicase activity. These additional results are now displayed in new Supplementary Figure 6b, listed in Supplementary Table 2 and briefly described in the revised text (line 333):

To test if ALKBH3 also modulates ASCC3-mediated DNA unwinding, we repeated stopped-flow/fluorescence-based DNA unwinding assays with ASCC3^{HR} in the presence of ALKBH3 and in the presence of a DNA trap. ALKBH3 did not significantly alter the amplitudes or rates of the fast and slow phases nor the amplitude-weighted unwinding rate constant of the biphasic unwinding reaction (Supplementary Fig. 6b; Supplementary Table 2).

New Supplementary Figure 6b:

Minor:

The authors mapped the interacting regions by SEC. Is there a reason why the SEC profiles are not shown?

There is no special reason why the SEC profiles are not shown. However, given that we work with purified proteins, we believe that the SDS-PAGE analyses of the elution fractions adequately represent the interactions or lack of and that the elution profiles would add little additional information. We would therefore prefer to stick to the presentations limited to the SDS-PAGE analyses to keep the respective figures concise.

It would be great if in Figure 4B the WT could be plotted again so that one can visually inspect the difference in the point mutants.

We agree and have now included reference traces in the relevant figure panels. To clearly mark the respective traces as references, we included the respective curve fits as dashed traces. Here new Figure 3b as an example:

In Figure 5a, I have problems seeing the path for the DNA. The figure might suffer from the PDF conversion, but also, the electrostatic surface potential makes it challenging to see.

We again agree and have decided to remove this panel, as the situation is also illustrated in the former Figure 5d (now new Figure 4a). Here the new Figure 4a:

The ssDNA region from the Hel308 structure only features 9 nucleotides – in figure 5a 24 nts, how was this modeled?

We are sorry that we have described the modeling too briefly in the first version of the manuscript. We now have added a more detailed description (line 279):

We thus modeled a putative path of ssDNA through ASCC3^{HR} by superimposing a structure of the Hel308 DNA helicase in complex with DNA (PDB ID 2P6R)⁴⁰ on both ASCC3^{HR} cassettes, transferring the Hel308 DNA substrates to both cassettes of the ASCC3^{HR} model and removing the dsDNA portion from the DNA transferred to the CC (Fig. 4a). In the resulting model, the ssDNA region transferred to the NC could be pseudo-continuous with the ssDNA region transferred to the CC, indicating that a longer ssDNA could be threaded consecutively through both helicase cassettes and might exit the CC close to the TRIP4 ASCH domain (Fig. 4a).

Figure 5b: DNA T24 is a defined band but for T36 and T48 a band resembling T24 is also present. Are the DNAs degraded or?

For the reviewer's information, here is a gel image showing the same experiment but including DNA samples without UV irradiation:

Comparing the longer DNAs with and without UV irradiation (red boxes) shows that the bands representing short oligos are already present in the non-irradiated samples, albeit at lower abundance. Thus, these bands most likely represent a mixture of truncated synthetic products, DNA degraded during labeling and DNA degraded upon UV irradiation. We now mention the likely source of these bands in the corresponding figure legend (line 1029):

Asterisks, bands in T_{24} , T_{36} and T_{48} samples representing truncated synthetic products and DNA degraded during labeling and/or UV irradiation.

In the results section, the MMS assay is not explained – it might be worth adding what MMS does and how this connects to ALKBH3.

We agree and have added a corresponding short explanation (line 340):

To this end, we knocked out TRIP4 via CRISPR/Cas9-based genome engineering in human PC-3 cells (Fig. 5b) and tested the response of the edited and parental cells to methyl methanesulfonate (MMS) that elicits DNA alkylation damage by methylating deoxyguanine (N7) and deoxyadenine (N3).

Reviewer #2:

The manuscript “Extended DNA threading through a dual-engine motor module in the activating signal co-integrator complex” by Jia, et al, describes the structure and activity of the ASCC3 subunit of the human activating signal co-integrator complex (ASCC). ASCC plays roles in transcriptional regulation, DNA repair and ribosome quality control. ASCC3 (the primary focus of this study) is a large Ski2-like DNA helicase/translocase that contains 2 helicase cassettes (an N-terminal cassette and a C-terminal cassette). In other characterized Ski2-like tandem-cassette helicases, only the N-terminal cassette exhibits helicase activity. However, in this study, the authors demonstrate that both cassettes of ASCC3 are functional. A cryo-EM structure shows how the cassettes pack together. Modeling of a DNA substrate onto the structure, combined with cross-linking/mass spectroscopy analysis, suggests that the DNA moves directly from the N-terminal to the C-terminal cassette.

Additionally, ASCC1, an activator of ASCC3 helicase activity, is shown to interact directly with the C-terminal cassette of ASCC3 through the ZnF domain using a combination of deletion mutants and analytical size-exclusion chromatography. The binding interaction is captured in the cryo-EM structure.

This is a thorough and well-written manuscript that makes an important contribution to the field. The observation that both cassettes are functional is particularly interesting and will be of interest to the broader helicase community. I have no concerns with the substance of the manuscript. However, the authors could consider the following comments:

We thank the reviewer for the very positive overall evaluation of our work, in particular for considering our manuscript to be well-written and for regarding our results as an important contribution.

It is noted in the Discussion that ASCC3 exhibits unwinding activity on RNA substrates, although at a significantly reduced level compared to DNA (data not shown). Can the authors provide data to demonstrate how activity compares between RNA and DNA?

We have now included data on ASCC3^{HR}-mediated and ASCC3^{HR}-TRIP4-mediated unwinding of RNA to the revised manuscript. The additional results are displayed in new Figure 3d and described in the revised text (line 249):

ASCC3 exhibits TRIP4-modulated RNA unwinding activity

As ASCC3 is thought to translocate RNA during ribosome quality control, we also tested whether ASCC3^{HR} can unwind RNA duplexes and whether this activity is likewise modulated by TRIP4. Using an RNA duplex of analogous sequence as the employed DNA substrate, stopped-flow/fluorescence-based unwinding assays in the absence of a trap revealed that ASCC3 indeed also exhibits TRIP4-modulated RNA unwinding activity (Fig. 3d). Again, biphasic time traces were observed, but the influence of TRIP4 on ASCC3^{HR}-mediated RNA unwinding was complex. In the presence of TRIP4, the rate constant for the fast phase was decreased while the rate constant for the slow phase was enhanced; at the same time, TRIP4 led to a ~ 10-fold increased amplitude for the fast component (Supplementary Table 2). Thus, while the amplitude-weighted RNA unwinding rate constant of ASCC3^{HR} ($k_{uaw} = 0.073 \text{ s}^{-1}$) was reduced in the presence of TRIP4 ($k_{uaw} = 0.046 \text{ s}^{-1}$), a larger fraction of ASCC3^{HR} molecules productively engaged the RNA substrate in the presence of TRIP4, leading to a larger fraction of RNA duplexes being unwound.

New Figure 3d:

Fig 1. While I have no concerns with the conclusions presented by the authors, the layout of the cropped gels in Fig 1 are somewhat disorienting – specifically when the width of the gels differs in a single panel (e.g. panel b). Additionally, the cropping of the final two gels in panel e appears to be a bit aggressive. The full gels included as source data provide essential and reassuring context. I don't have specific suggestions for how to improve the figure, but I encourage the authors to consider adjustments to improve readability.

We apologize for the sub-optimal display. Taking also the advice from another reviewer, we have moved most of the former Figure 1 panels to the Supplement (new Supplementary Figure 1). We have adjusted the panels to better highlight the alignment of lanes by illustrating missing lanes in some gels by dashed boxes. Given the original gels in the Source Data file, we hope that the presentation is now acceptable.

Fig 5. Panel a and d. It isn't clear what the boxed rotations refer to. My guess is that it refers to rotations compared to Fig 2, but I don't see it specified in the figure or the figure legend. Also, "orthogonal" (pg 40 ln 917) generally refers to right angles. I think the figure is showing a 180 deg rotation instead.

The rotation symbols do indeed indicate the views of structural figures relative to the overviews shown in Figure 1c/d, left panels (former Figure 2a/b, left). In the original manuscript, this was stated in the legend to the former Figure 2a (now Figure 1c). We now have added corresponding statements to all relevant figures:

Rotation symbol(s), orientation(s) relative to Fig. 1c,d, left panels.

The reviewer is correct that "orthogonal" was the wrong description and we apologize for the mistake. Due to a point raised by reviewer 1, we have now omitted the panel in question.

Minor comments:

Pg 12 ln 317. Remove comma after "mechanism"

Done.

Pg 13 ln 359. Remove comma after "possible"

Done.

Pg 16 In 434. Should be "pre-equilibrated"

Corrected.

Reviewer #3:

Jia J. et al report the structural and biochemical characterization of the activating signal co-integrator complex (ASCC) subunit 3 in complex with ASC1. The authors show that both helicase domains of ASCC3 are active in contrast to the homologue Ski2-like helicase SNRNP200. The data further indicate that ASC1 and ALKBH3 binding to ASCC3 is mutually exclusive. Although the structure of ASCC3 in complex with ASCC2 has been solved before by the authors, the interaction with the regulatory ASC1 subunit was unknown before and gives important insight into the regulation of the ASCC3 helicase. The data are detailed and solid and proof all conclusions drawn. However, I have some suggestion to improve the manuscript.

We thank the reviewer for the positive general assessment of our manuscript and for considering our data as detailed, solid and in support of the conclusions drawn.

For readers from outside the field it is difficult to understand which ASCC proteins form a stable complex, and which are auxiliary proteins. For example, are ASCC1 and ASCC2 constitutively bound to ASCC3? Where would ASCC2 interact in the ASCC3-ASC1 complex or is the interaction also mutual exclusive? Maybe the comparison of the ASCC3-ASCC2 and ASCC3-ASC1 complex would help to understand the regulation?

We thank the reviewer for pointing this out. Indeed, considering all functions that ASCC has been implicated in it is difficult to distinguish core subunits and auxiliary proteins. We admit that our original introduction into this issue may have been confusing. Indeed, only ASCC2 and ASCC3 have so far been implicated in all of the major relevant functions, *i.e.* transcriptional regulation, DNA alkylation damage repair and ribosome quality control. Depending on the process, additional subunits or “auxiliary proteins” join ASCC2-ASCC3. We have thoroughly revised the introduction, avoiding the distinction of “core” and “auxiliary” subunits.

We had previously shown that ASCC2 interacts with a small helical domain of the ASCC3 N-terminal region but not with the ASCC3 helicase region (as TRIP4 does), elucidated the structural basis of the ASCC2-ASCC3 interaction and studied its relevance for ASCC3 auto-regulation (Jia *et al.*, *Nat Commun*, 2020). While we managed to obtain a ternary ASCC2-ASCC3-TRIP4 complex from recombinant components, we did not obtain a cryoEM reconstruction in which we could visualize all three components. Possibly the ASCC2-bound N-terminal region of ASCC3 remains flexible and multiply oriented relative to the TRIP4-bound helicase region in the isolated ternary complex. We also did not detect a direct, stable ASCC2-TRIP4 interaction *in vitro*. We therefore focused on the ASCC3^{HR}-TRIP4 interaction and its functional consequences for ASCC3 helicase activity in the present manuscript.

Therefore, we now also briefly recapitulate the mode of interaction of ASCC2 and ASCC3 at the end of the introduction to contrast it with the ASCC3-TRIP4 interaction investigated here, but in the following focus on the ASCC3-TRIP4 interaction (line 105):

Previously, we showed that ASCC2 associates with a small helical domain in the N-terminal region of ASCC3, an interaction that may auto-regulate ASCC3.¹⁰ Here, we report that the TRIP4 protein binds the ASCC3 helicase region and supports a hitherto unobserved mechanism of nucleic acid translocation/unwinding.

1) Page 2 Line 29: What does it mean that ASCC3 is an unconventional helicase?

We avoided the term “unconventional” in the revised manuscript and now explicitly state that, apart from SNRNP200, ASCC3 is the only known helicase with dual Ski2-like helicase units in human.

2) Page 3, line 55-60: The sentence is too long and barely understandable for people outside the field.

Due to a comment by reviewer 1, we thoroughly revised the introduction, leaving out information not directly important for the present manuscript. In the course of revising the introduction, we omitted the sentence in question.

3) It would be better to put some of the western blots in Figure 1 (c, e, f) in the supplementary data and show the structures from figure 2a and b in Figure 1 instead. This is necessary to immediately understand the color code in Fig. 2 a,b) Figure 2 c has a completely different color code and Figure 2d fits to a and b again. This is a bit confusing. Further the numbering and the interacting residues in Figure 2d (first image) are barely visible.

We thank the reviewer for pointing out these deficits in our original figure and for suggestions on how to improve it, which we implemented. First, in the revised main text Figure 1, we now only show the interaction of ASCC3^{HR} with full-length TRIP4 (formerly ASC1) and a TRIP4 fragment representing the portion of the protein visible in our cryoEM reconstruction (panel b). All other SDS-PAGE gels monitoring interactions among other protein regions have been moved to the Supplement (new Supplementary Figure 1). Second, we now display the protein schemes (panel a), the main interaction tests (panel b) and the main structural figures (panels c-g) together in the new Figure 1. Third, we have adjusted the color coding and tried to improve labeling. In domain schemes and structure figures, we now display relevant ASCC3^{HR} regions in different shades of gray and TRIP4 uniformly in red (as ZnF, lasso and ASCH regions are well separated and discernible in the structural overviews). Fourth, we removed the former Figure 2c from the main text figure to avoid confusion due to different color coding. We now show the corresponding information in the new Supplementary Figure 4, in which we highlight ASCC3^{HR} domains and similarity to SNRNP200^{HR} by coloring domains blue to red from N-terminus to C-terminus within each ASCC3^{HR}/SNRNP200^{HR} helicase cassette. New Figure 1:

New Supplementary Figure 4:

a ASCC3^{HR}**b** SNRNP200^{HR}
- 4) *The information content of the DNA binding model in Figure 5a is rather low and it somehow looks like the DNA is bound by negatively charged residues. Why is the DNA chain broken in Figure 5d? Could you please optimize this model or illustrate it better?*

We apologize for the suboptimal presentation. As a similar point was raised also by reviewer 1, we have decided to remove the panel in question, as the situation is also illustrated in the former Figure 5d (now new Figure 4a). The DNA was broken, as we had initially simply transferred single-stranded DNA regions from the Hel308-DNA complex to the NC and CC of ASCC3^{HR} separately by superposition of the proteins without connecting the DNA regions. However, after the superposition, the single-stranded DNA regions were pseudo-continuous, such that they could be connected without further adjustments. We now display the modeled regions of bound single-stranded DNA as a continuous strand. Here the new panel Figure 4a after the adjustments:

- 5) Just out of curiosity: The protein-DNA crosslink looks very efficient - what was the problem with the cryo EM structure of the DNA bound complex. Did you try different grid types?

Indeed, we have extensively screened different parameters to obtain a cryoEM structure of an ASCC3^{HR}-TRIP4-DNA complex, including grid surfaces (e.g. graphene as a support) as well as various buffer conditions. Apparently, the complex disintegrates during vitrification, perhaps due to fast binding/dissociation kinetics.

- 6) Fig. 3c (Input) and Fig. 5b some western blots look like modified in contrast and brightness too much. Could you please provide an unmodified image?

Scans of any gel shown have only been uniformly adjusted for brightness/contrast. Original images of all gels are contained in the Source Data file.

REVIEWERS' COMMENTS

Reviewer #1 (Remarks to the Author):

The introduction reads very well now and nicely prepares the reader for the results/discussion section. By changing/improving the figures, they are now simpler and more intuitive to understand. The flow has been improved and the text modification help the reader in understanding/interpreting the experiments.

The inclusion of DNA binding experiments is great and makes the manuscript stronger. Also, the fact that ALKBH3 does not stimulate the helicase in contrast to TRIP4 is very interesting. Moreover, I also like the inclusion of the dsRNA unwinding experiments.

Minor

The authors used MST and refer to it as multi-scale thermophoresis in main text but microscale thermophoresis in the methods section. I assume microscale thermophoresis is the correct wording.

Overall, the authors did a great job in improving the manuscript and I enjoyed reading the new version. I recommend publication.

Reviewer #2 (Remarks to the Author):

The authors have appropriately addressed all the concerns from my previous review.

Reviewer #3 (Remarks to the Author):

The authors have addressed all points appropriately and optimized the manuscript as requested. I'm happy to support the revised manuscript for acceptance.

Response to Reviewer Comments

Reviewer comments are repeated in bold italics, responses are in regular font.

Reviewer #1:

The introduction reads very well now and nicely prepares the reader for the results/discussion section. By changing/improving the figures, they are now simpler and more intuitive to understand. The flow has been improved and the text modification help the reader in understanding/interpreting the experiments. The inclusion of DNA binding experiments is great and makes the manuscript stronger. Also, the fact that ALKBH3 does not stimulate the helicase in contrast to TRIP4 is very interesting. Moreover, I also like the inclusion of the dsRNA unwinding experiments.

We thank the reviewer for these positive comments and for appreciating the revision of our manuscript.

Minor:

The authors used MST and refer to it as multi-scale thermophoresis in main text but microscale thermophoresis in the methods section. I assume microscale thermophoresis is the correct wording.

We thanks the reviewer for catching this typo. Indeed “microscale” thermophoresis is correct. We corrected the error.

Overall, the authors did a great job in improving the manuscript and I enjoyed reading the new version. I recommend publication.

We thank the reviewer for recommending publication of our manuscript.

Reviewer #2:

The authors have appropriately addressed all the concerns from my previous review.

We thank the reviewer for the positive evaluation.

Reviewer #3:

The authors have addressed all points appropriately and optimized the manuscript as requested. I'm happy to support the revised manuscript for acceptance.

We thank the reviewer for the positive evaluation and for supporting acceptance of the revised manuscript.